# On the Efficient Implementation of High Accuracy Optimality of Profile Maximum Likelihood

**Moses Charikar**
Stanford University
moses@cs.stanford.edu

**Zhihao Jiang**
Stanford University
faebdc@stanford.edu

**Kirankumar Shiragur**
Stanford University
shiragur@stanford.edu

**Aaron Sidford**
Stanford University
sidford@stanford.edu

## Abstract

We provide an efficient unified plug-in approach for estimating symmetric properties of distributions given $n$ independent samples. Our estimator is based on profile-maximum-likelihood (PML) and is sample optimal for estimating various symmetric properties when the estimation error $\epsilon \gg n^{-1/3}$. This result improves upon the previous best accuracy threshold of $\epsilon \gg n^{-1/4}$ achievable by polynomial time computable PML-based universal estimators [ACSS21, ACSS20]. Our estimator reaches a theoretical limit for universal symmetric property estimation as [Han21] shows that a broad class of universal estimators (containing many well known approaches including ours) cannot be sample optimal for every 1-Lipschitz property when $\epsilon \ll n^{-1/3}$.

## 1 Introduction

Given $n$ independent samples $y_1, ..., y_n \in \mathcal{D}$ from an unknown discrete distribution $\mathbf{p} \in \Delta^{\mathcal{D}}$ the problem of estimating properties of $\mathbf{p}$, e.g. entropy, distance to uniformity, support size and coverage are among the most fundamental in statistics and learning. Further, the problem of estimating *symmetric properties* of distributions $\mathbf{p}$ (i.e. properties invariant to label permutations) are well studied and have numerous applications [Cha84, BF93, CCG$^+$12, TE87, Für05, KLR99, PBG$^+$01, DS13, RCS$^+$09, GTPB07, HHRB01].

Over the past decade, symmetric property estimation has been studied extensively and there have been many improvements to the time and sample complexity for estimating different properties, e.g. support [VV11b, WY15], coverage [ZVV$^+$16, OSW16], entropy [VV11b, WY16, JVHW15], and distance to uniformity [VV11a, JHW16]. Towards unifying the attainment of computationally-efficient, sample-optimal estimators a striking work of [ADOS17] provided a universal plug-in approach based on a (approximate) profile maximum likelihood (PML) distribution, that (approximately) maximizes the likelihood of the observed profile (i.e. multiset of observed frequencies).

Formally, [ADOS17] showed that given $y_1, ..., y_n$ if there exists an estimator for a symmetric property $f$ achieving accuracy $\epsilon$ and failure probability $\delta$, then this PML-based plug-in approach achieves error $2\epsilon$ with failure probability $\delta \exp(3\sqrt{n})$. As the failure probability $\delta$ for many estimators for well-known properties (e.g. support size and coverage, entropy, and distance to uniformity) is roughly $\exp(-\epsilon^2 n)$, this result implied a sample optimal unified approach for estimating these properties when the estimation error $\epsilon \gg n^{-1/4}$.

36th Conference on Neural Information Processing Systems (NeurIPS 2022).

This result of [ADOS17] laid the groundwork for a line of work on the study of computational and statistical aspects of PML-based approaches to symmetric property estimation. For example, follow up work of [HS21] improved the analysis of [ADOS17] and showed that the failure probability of PML is at most $\delta^{1-c}\exp(-n^{1/3+c})$, for any constant $c > 0$ and therefore it is sample optimal in the regime $\epsilon \gg n^{-1/3}$. The condition $\epsilon \gg n^{-1/3}$ on the optimality of PML is tight [Han21], in the sense that, PML is known to be not sample optimal in the regime $\epsilon \ll n^{-1/3}$. In fact, no estimator (that obeys some mild conditions), is sample optimal for estimating all symmetric properties in the regime $\epsilon \ll n^{-1/3}$; see Section 2 after Theorem 2.6 for details.

We also remark that the statistical guarantees in [ADOS17, HS21] hold for any $\beta$-approximate PML[1] for suitable values of $\beta$. In particular, [HS21] showed that any $\beta$-approximate PML for $\beta > \exp(-n^{1-c'})$ and any constant $c' > 0$, has a failure probability of $\delta^{1-c}\exp(n^{1/3+c} + n^{1-c'})$ for any constant $c > 0$. These results further imply a sample optimal estimator in the regime $\epsilon \gg n^{-\min(1/3,c'/2)}$ for properties with failure probability less than $\exp(-\epsilon^2 n)$. Note that better approximation leads to a larger range of $\epsilon$ for which the estimator is sample optimal.

Regarding computational aspects of PML, [CSS19a] provided the first efficient algorithm with a non-trivial approximation guarantee of $\exp(-n^{2/3}\log n)$, which further implied a sample optimal universal estimator for $\epsilon \gg n^{-1/6}$. This was then improved by [ACSS21] which showed how to efficiently compute PML to higher accuracy of $\exp(-\sqrt{n}\log n)$ thereby achieving a sample optimal universal estimator in the regime $\epsilon \gg n^{-1/4}$. The current best polynomial time approximate PML algorithm by [ACSS20] achieves an accuracy of $\exp(-k\log n)$, where $k$ is the number of distinct observed frequencies. Although this result achieves better instance based statistical guarantees, in the worst case it still only implies a sample optimal universal estimator in the regime $\epsilon \gg n^{-1/4}$.

In light of these results, a key open problem is to close the gap between the regimes $\epsilon \gg n^{-1/3}$ and $\epsilon \gg n^{-1/4}$, where the former is the regime in which PML based estimators are statistically optimal and the later is the regime where efficient PML based estimators exist. In this work we ask:

> *Is there an efficient approximate PML-based estimator that is sample optimal for $\epsilon \gg n^{-1/3}$.*

In this paper, we answer this question in the affirmative. In particular, we give an efficient PML-based estimator that has failure probability at most $\delta^{1-c}\exp(n^{1/3+c} + n^{1-c'})$, and consequently is sample optimal in the regime $\epsilon \gg n^{-1/3}$. As remarked, this result is tight in the sense that PML and a broad class of estimators are known to be not optimal in the regime $\epsilon \ll n^{-1/3}$.

To obtain this result we depart slightly from the previous approaches in [ADOS17, CSS19a, ACSS21]. Rather than directly compute an approximate PML distribution we compute a weaker notion of approximation which we show suffices to get us the desired universal estimator. We propose a notion of a $\beta$-*weak approximate PML distribution* inspired by [HS21] and show that an $\exp(-n^{1/3}\log n)$-weak approximate PML achieves the desired failure probability of $\delta^{1-c}\exp(n^{1/3+c})$ for any constant $c > 0$. Further, we provide an efficient algorithm to compute an $\exp(-n^{1/3}\log n)$-weak approximate PML distribution. Our paper can be viewed as an efficient algorithmic instantiation of [HS21].

Ultimately, our algorithms use the convex relaxation presented in [CSS19a, ACSS21] and provide a new rounding algorithm. We differ from the previous best $\exp(-k\log n)$ approximate PML algorithm [ACSS20] only in the matrix rounding procedure which controls the approximation guarantee. At a high level, the approximation guarantee for the rounding procedure in [ACSS20] is exponential in the sum of matrix dimensions. In the present work, we need to round a rectangular matrix with an approximation exponential in the smaller dimension, which may be infeasible for arbitrary matrices. Our key technical innovation is to introduce a *swap* operation (see Section 4.1) which facilitates such an approximation guarantee. In addition to a better approximation guarantee than [ACSS20], our algorithm also exhibits better run times (see Section 2).

**Organization:** We introduce preliminaries in Section 1.1. In Section 2, we state our main results and also cover related work. In Section 3, we provide the convex relaxation to PML studied in [CSS19a, ACSS21]. Finally, in Section 4, we provide a proof sketch of our main computational result. Many proofs are then differed to the appendix.

---

[1]$\beta$-approximate PML is a distribution that achieves a multiplicative $\beta$-approximation to the PML objective.

## 1.1 Preliminaries

**General notation:** For matrices $\mathbf{A}, \mathbf{B} \in \mathbb{R}^{s \times t}$, we use $\mathbf{A} \leq \mathbf{B}$ to denote that $\mathbf{A}_{ij} \leq \mathbf{B}_{ij}$ for all $i \in [s]$ and $j \in [t]$. We let $[a, b]$ and $[a, b]_{\mathbb{R}}$ denote the interval $\geq a$ and $\leq b$ of integers and reals respectively. We use $\widetilde{O}(\cdot), \widetilde{\Omega}(\cdot)$ notation to hide all polylogarithmic factors in $n$ and $N$. We let $a_n \gg b_n$ to denote that $a_n \in \Omega(b_n n^c)$ or $b_n \in O(n^{-c} a_n)$, for some small constant $c > 0$.

Throughout this paper, we assume we receive a sequence of $n$ independent samples from a distribution $\mathbf{p} \in \Delta^{\mathcal{D}}$, where $\Delta^{\mathcal{D}} \stackrel{\text{def}}{=} \{\mathbf{q} \in [0, 1]_{\mathbb{R}}^{\mathcal{D}} \mid \|q\|_1 = 1\}$ is the set of all discrete distributions supported on domain $\mathcal{D}$. Let $\mathcal{D}^n$ be the set of all length $n$ sequences of elements of $\mathcal{D}$ and for $y^n \in \mathcal{D}^n$ let $y_i^n$ denoting its $i$th element. Let $\mathbf{f}(y^n, x) \stackrel{\text{def}}{=} |\{i \in [n] \mid y_i^n = x\}|$ and $\mathbf{p}_x$ be the frequency and probability of $x \in \mathcal{D}$ respectively. For a sequence $y^n \in \mathcal{D}^n$, let $\mathbf{M} = \{\mathbf{f}(y^n, x)\}_{x \in \mathcal{D}} \setminus \{0\}$ be the set of all its non-zero distinct frequencies and $\mathbf{m}_1, \mathbf{m}_2, \ldots, \mathbf{m}_{|\mathbf{M}|}$ be these distinct frequencies. The *profile* of a sequence $y^n$ denoted $\phi = \Phi(y^n)$ is a vector in $\mathbb{Z}^{|\mathbf{M}|}$, where $\phi_j \stackrel{\text{def}}{=} |\{x \in \mathcal{D} \mid \mathbf{f}(y^n, x) = \mathbf{m}_j\}|$ is the number of domain elements with frequency $\mathbf{m}_j$. We call $n$ the length of profile $\phi$ and let $\Phi^n$ denote the set of all profiles of length $n$. The probability of observing sequence $y^n$ and profile $\phi$ for distribution $\mathbf{p}$ are $\mathbb{P}(\mathbf{p}, y^n) = \prod_{x \in \mathcal{D}} \mathbf{p}_x^{\mathbf{f}(y^n, x)}$ and $\mathbb{P}(\mathbf{p}, \phi) = \sum_{\{y^n \in \mathcal{D}^n \mid \Phi(y^n) = \phi\}} \mathbb{P}(\mathbf{p}, y^n)$.

**Profile maximum likelihood:** A distribution $\mathbf{p}_\phi \in \Delta^{\mathcal{D}}$ is a *profile maximum likelihood* (PML) distribution for profile $\phi \in \Phi^n$ if $\mathbf{p}_\phi \in \operatorname{argmax}_{\mathbf{p} \in \Delta^{\mathcal{D}}} \mathbb{P}(\mathbf{p}, \phi)$. Further, a distribution $\mathbf{p}_\phi^\beta$ is a $\beta$-*approximate PML* distribution if $\mathbb{P}(\mathbf{p}_\phi^\beta, \phi) \geq \beta \cdot \mathbb{P}(\mathbf{p}_\phi, \phi)$. For a distribution $\mathbf{p}$ and a length $n$, let $\mathbf{X}$ be a random variable that takes value $\phi \in \Phi^n$ with probability $\mathbb{P}(\mathbf{p}, \phi)$. We call $H(\mathbf{X})$ (entropy of $\mathbf{X}$) the *profile entropy* with respect to $(\mathbf{p}, n)$ and denote it by $H(\Phi^n, \mathbf{p})$.

**Probability discretization:** Let $\mathbf{R} \stackrel{\text{def}}{=} \{\mathbf{r}_i\}_{i \in [1, \ell]}$ be a finite discretization of the probability space, where $\mathbf{r}_i \in [0, 1]_{\mathbb{R}}$ and $\ell \stackrel{\text{def}}{=} |\mathbf{R}|$. We call $\mathbf{q} \in [0, 1]_{\mathbb{R}}^{\mathcal{D}}$ a *pseudo-distribution* if $\|\mathbf{q}\|_1 \leq 1$ and a *discrete pseudo-distribution* with respect to $\mathbf{R}$ if all its entries are in $\mathbf{R}$ as well. We use $\Delta_{\text{pseudo}}^{\mathcal{D}}$ and $\Delta_{\mathbf{R}}^{\mathcal{D}}$ to denote the set of all pseudo-distributions and discrete pseudo-distributions with respect to $\mathbf{R}$ respectively. In our work, we use the following most commonly used [CSS19a, ACSS21, ACSS20] probability discretization set. For any $\alpha > 0$,

$$\mathbf{R}_{n, \alpha} \stackrel{\text{def}}{=} \{1\} \cup \left\{ \frac{1}{2n^2} (1 + n^{-\alpha})^i \mid \text{for all } i \in \mathbb{Z}_{\geq 0} \text{ such that } \frac{1}{2n^2} (1 + n^{-\alpha})^i \leq 1 \right\}. \quad (1)$$

For all probability terms defined involving distributions $\mathbf{p}$, we extend those definitions to pseudo distributions $\mathbf{q}$ by replacing $\mathbf{p}_x$ with $\mathbf{q}_x$ everywhere.

See **??** for the definition of an estimator and optimal sample complexity.

## 2 Results

Here we provide our main results. In our first result (Theorem 2.2), we show that a weaker notion of approximate PML suffices to obtain the desired universal estimator. Later we show that these weaker approximate PML distributions can be efficiently computed (Theorem 2.3).

**Definition 2.1.** Given a profile $\phi$, we call a distribution $\mathbf{p}' \in \Delta^{\mathcal{D}}$ $\beta$-*approximate PML distribution with respect to $\mathbf{R}$ if* $\mathbb{P}(\mathbf{p}', \phi) \geq \beta \cdot \max_{\mathbf{q} \in \Delta_{\mathbf{R}}^{\mathcal{D}}} \mathbb{P}\left(\frac{\mathbf{q}}{\|\mathbf{q}\|_1}, \phi\right)$.

The above definition generalizes $\beta$-approximate PML distributions which is simply the special case when $\mathbf{R} = [0, 1]_{\mathbb{R}}$. Using our new definition, we show that for a specific choice of the discretization set $\mathbf{R}_{n, 1/3}$, a distribution $\mathbf{p}'$ that is an approximate PML with respect to $\mathbf{R}_{n, 1/3}$ suffices to obtain a universal estimator; this result is formally stated below.

**Theorem 2.2** (Competitiveness of an approximate PML w.r.t $\mathbf{R}$). *For symmetric property $f$, suppose there exists an estimator $\widehat{f}$ that takes input a profile $\phi \in \Phi^n$ drawn from $\mathbf{p} \in \Delta^{\mathcal{D}}$ and satisfies,*

$$\mathbb{P}\left(|f(\mathbf{p}) - \widehat{f}(\phi)| \geq \epsilon\right) \leq \delta,$$

then for $\boldsymbol{R} = \boldsymbol{R}_{n,1/3}$ *(See Equation* (1)*), a discrete pseudo distribution* $\boldsymbol{q}' \in \Delta_{\boldsymbol{R}}^{\mathcal{D}}$ *such that* $\boldsymbol{q}'/\|\boldsymbol{q}'\|_1$ *is an* $\exp(-O(|\boldsymbol{R}| \log n))$*-approximate PML distribution with respect to the* $\boldsymbol{R}$ *satisfies,*

$$\mathbb{P}\left(\left|f\left(\frac{\boldsymbol{q}'}{\|\boldsymbol{q}'\|_1}\right) - f(\boldsymbol{p})\right| \geq 2\epsilon\right) \leq \delta^{1-c} \exp(O(n^{1/3+c})), \quad \text{for any constant } c > 0 \ . \quad (2)$$

The proof of the above theorem is implicit in the analysis of [HS21], however we provide a short simpler proof using their continuity lemma (Lemma 2 in [HS21]). Note that the bound on the failure probability we get is the same asymptotically as that of exact PML from [HS21], which is known to be tight [Han21]. Furthermore, to achieve such an improved failure probability bound all we need is an approximate PML distribution with respect to $\boldsymbol{R}$, for some $\boldsymbol{R}$ which is of small size. Taking advantage of this fact and building upon [CSS19a, ACSS21], we provide a new rounding algorithm that outputs the desired approximate PML distribution with respect to $\boldsymbol{R}$.

**Theorem 2.3** (Computation of an approximate PML w.r.t $\boldsymbol{R}$). *We provide an algorithm that given a probability discretization set* $\boldsymbol{R} = \boldsymbol{R}_{n,\alpha}$ *for* $\alpha > 0$ *(See Equation* (1)*) and a profile* $\phi$ *with* $k$ *distinct frequencies, runs in time* $\widetilde{O}\left(|\boldsymbol{R}| + \frac{n}{\min(k,|\boldsymbol{R}|)}\left(\min(|\boldsymbol{R}|, n/k)k^\omega + \min(|\boldsymbol{R}|, k)k^2\right)\right)$, *where* $\omega < 2.373$ *is the current matrix multiplication constant [Wil12, Gal14, AW21] and returns a pseudo distribution* $\boldsymbol{q}' \in \Delta_{\boldsymbol{R}}^{\mathcal{D}}$ *such that,*

$$\mathbb{P}\left(\frac{\boldsymbol{q}'}{\|\boldsymbol{q}'\|_1}, \phi\right) \geq \exp\left(-O(\min(k, |\boldsymbol{R}|) \log n)\right) \cdot \max_{\boldsymbol{q} \in \Delta_{\boldsymbol{R}}^{\mathcal{D}}} \mathbb{P}\left(\frac{\boldsymbol{q}}{\|\boldsymbol{q}\|_1}, \phi\right) \ .$$

When $\boldsymbol{R} = \boldsymbol{R}_{n,1}$, our algorithm computes an $\exp(-O(k \log n))$ approximate PML distribution, therefore our result is at least as good as the previous best known approximate PML algorithm due to [ACSS20]. In comparison to [ACSS20], our rounding algorithm is simpler and we suspect, more practical. We provide a more detailed comparison to it later in this section.

**Applications:** Our main results have several applications which we discuss here. First note that, combining Theorem 2.2 and 2.3 immediately yields the following corollary.

**Corollary 2.4** (Efficient unified estimator). *Given a profile* $\phi \in \Phi^n$ *with* $k$ *distinct frequencies, we can compute an approximate PML distribution* $\boldsymbol{q}'$ *that satisfies Equation* (2) *in Theorem 2.2 in time* $\widetilde{O}\left(\frac{n}{\min(k,n^{1/3})}\left(\min(n^{1/3}, n/k)k^\omega + \min(n^{1/3}, k)k^2\right)\right)$.

For many symmetric properties the failure probability is exponentially small as stated below.

**Lemma 2.5** (Lemma 2 in [ADOS17], Theorem 3 in [HS21]). *For distance to uniformity, entropy, support size and coverage, and sorted* $\ell_1$ *distance there exists an estimator that is sample optimal and the failure probability is at most* $\exp(-\epsilon^2 n^{1-\alpha})$ *for any constant* $\alpha > 0$.

The above result combined with Corollary 2.4, immediately yields the following theorem.

**Theorem 2.6** (Efficient sample optimal unified estimator). *There exists an efficient approximate PML-based estimator that for* $\epsilon \gg n^{-1/3}$ *and symmetric properties such as, distance to uniformity, entropy, support size and coverage, and sorted* $\ell_1$ *distance achieves optimal sample complexity and has failure probability upper bounded by* $\exp(-n^{1/3})$.

As our work computes an $\exp(-O(k \log n))$ approximate PML, we recover efficient version of Lemma 2.3 and Theorem 2.4 from [ACSS20]. The first result uses $\exp(-O(k \log n))$ approximate PML algorithm to efficiently implement an estimator that has better statistical guarantees based on profile entropy [HO20] (See Section 1.1). The second result provides an efficient implementation of the PseudoPML estimators [CSS19b, HO19]. Please refer to the respective papers for further details.

**Tightness of our result:** Recall that [HS21] showed that the failure probability of an (approximate) PML based estimator is upper bounded by $\delta^{1-c} \exp(-n^{1/3+c})$, for any constant $c > 0$. This result further implied a sample optimal universal estimator in the regime $\epsilon \gg n^{-1/3}$ for various symmetric properties (Theorem 2.6). In our work, we efficiently recover these results and a natural question to ask here is if these results can be improved.

As remarked earlier, [Han21] showed that the condition for optimality of PML ($\epsilon \gg n^{-1/3}$) is in some sense tight. More formally, they showed that PML is not sample optimal in estimating every

1-Lipschitz property in the regime $\epsilon \ll n^{-1/3}$. In fact, the results in [Han21] hold more broadly for any universal plug-in based estimator that outputs a distribution $\hat{\mathbf{p}}$ satisfying,

$$\max_{\mathbf{p} \in \Delta^{\mathcal{D}}} \mathbb{E} \|\mathbf{p} - \hat{\mathbf{p}}\|_1^{\text{sorted}} \leq A(n) \sqrt{k/n} \,,$$

where $A(n) \leq n^\gamma$ for every $\gamma > 0$ and $\|\mathbf{p} - \mathbf{q}\|_1^{\text{sorted}} \overset{\text{def}}{=} \min_{\text{permutations } \sigma} \|\mathbf{p} - \mathbf{q}_\sigma\|_1$ denotes the sorted $\ell_1$ distance between $\mathbf{p}$ and $\mathbf{q}$. In other words, if an estimator is based on a reasonably good estimate of the true distribution $\mathbf{p}$ (in terms of sorted-$\ell_1$ distance), then it cannot be sample optimal for every 1-Lipschitz property. Furthermore, many well-known universal estimators including PML and LLM [HJW18] indeed provide a reasonably good estimate of the true distribution and therefore cannot be sample optimal in the regime $\epsilon \ll n^{-1/3}$. Please refer to [Han21] for further details.

**Comparison to approximate PML algorithms:** All prior provable approximate PML algorithms [CSS19a, ACSS21, ACSS20] have two key steps: (Step 1) solve a convex approximation to the PML and (Step 2) round the (fractional) solution to a valid approximate PML distribution.

A convex approximation to PML was first provided in [CSS19a] and a better analysis for it is shown in [ACSS21]. In particular, [CSS19a] and [ACSS21] showed that an integral optimal solution to step 1 approximates the PML up to accuracy $\exp(-n^{2/3} \log n)$ and $\exp(-\min(k, |\mathbf{R}|) \log n)$ respectively, where $k$ and $|\mathbf{R}|$ are the number of distinct frequencies and distinct probability values respectively. In addition to the loss from convex approximation, the previous algorithms also incurred a loss in the rounding step (Step 2). The loss in the rounding step for the previous works is bounded by $\exp(-n^{2/3} \log n)$ [CSS19a], $\exp(-\sqrt{n} \log n)$ [ACSS21] and $\exp(-k \log n)$ [ACSS20].

In our work, we show that there exists a choice of $\mathbf{R}$ (=$\mathbf{R}_{n,1/3}$) that is of small size ($|\mathbf{R}| \leq n^{1/3}$) and suffices to get the desired universal estimator. As $|\mathbf{R}| \leq n^{1/3}$, our approach only incur a loss of $\exp(-\min(k, |\mathbf{R}|) \log n) \in \exp(-n^{1/3} \log n)$ in the convex approximation step (Step 1). Furthermore for the rounding step (Step 2), we provide a new simpler and a practical rounding algorithm with a better approximation loss of $\exp(-O(\min(k, |\mathbf{R}|) \log n)) \in \exp(-O(n^{-1/3} \log n))$.

Regarding the run times, both [ACSS20] and ours have run times of the form $\mathcal{T}_{\text{solve}} + \mathcal{T}_{\text{sparsify}} + \mathcal{T}_{\text{round}}$, where the terms correspond to the time required to solve the convex program, sparsify and round a solution. In our algorithm, we pay the same cost as [ACSS20] for the first two steps but our run time guarantees are superior to theirs in the rounding step. In particular, the run time of [ACSS20] is shown as a large polynomial and perhaps not practical as their approach requires enumerating all the approximate min cuts. In contrast, our algorithm has a run time that is subquadratic.

**Other related work** PML was introduced by [OSS+04]. Many heuristic approaches have been proposed to compute approximate PML, such as the EM algorithm in [OSS+04], an algebraic approaches in [ADM+10], Bethe approximation in [Von12] and [Von14], and a dynamic programming approach in [PJW17]. For the broad applicability of PML in property testing and to estimate other symmetric properties please refer to [HO19]. Please refer to [HO20] for details related to profile entropy. Other approaches for designing universal estimators are: [VV11b] based on [ET76], [HJW18] based on local moment matching, and variants of PML by [CSS19b, HO19] that weakly depend on the target property that we wish to estimate. Optimal sample complexities for estimating many symmetric properties were also obtained by constructing property specific estimators, e.g. support [VV11b, WY15], support coverage [OSW16, ZVV+16], entropy [VV11b, WY16, JVHW15], distance to uniformity [VV11a, JHW16], sorted $\ell_1$ distance [VV11a, HJW18], Renyi entropy [AOST14, AOST17], KL divergence [BZLV16, HJW16] and others.

**Limitations of our work** One of the limitations of all the provable approximate PML algorithms [CSS19a, ACSS21, ACSS20] (including ours) is that they require the solution of a convex program that approximates the PML objective and all these previous works use the CVX solver which is not practical for large sample instances; note that our results hold for small error regimes which lead to such large sample instances. Therefore, designing a practical algorithm to solve the convex program is an important future research direction. As discussed above, local moment matching (LLM) based approach is another universal approach for property estimation. It is unclear which of the two (PML or LLM) can lead to practical algorithms.

# 3 Convex relaxation to PML

Here we restate the convex program from [CSS19a] that approximates the PML objective. The current best analysis of this convex program is in [ACSS21]. We first describe the notation and later state several results from [CSS19a, ACSS21] that capture the guarantees of the convex program.

**Notation:** For any matrices $\mathbf{X} \in \mathbb{R}^{a \times c}$ and $\mathbf{Y} \in \mathbb{R}^{b \times c}$, we let $\operatorname{concat}(\mathbf{X}, \mathbf{Y})$ denote the matrix $\mathbf{W} \in \mathbb{R}^{(a+b) \times c}$ that satisfies, $\mathbf{W}_{i,j} = \mathbf{X}_{i,j}$ for all $i \in [a]$ and $j \in [c]$ and $\mathbf{W}_{a+i,j} = \mathbf{Y}_{ij}$ for all $i \in [b]$ and $j \in [c]$. Recall we let $\mathbf{R} \stackrel{\text{def}}{=} \{\mathbf{r}_i\}_{i \in [\ell]}$ be a finite discretization of the probability space, where $\mathbf{r}_i \in [0,1]_{\mathbb{R}}$ and $\ell \stackrel{\text{def}}{=} |\mathbf{R}|$. Let $\mathbf{r} \in [0,1]_{\mathbb{R}}^{\ell}$ be a vector whose $i$'th element is equal to $\mathbf{r}_i$.

**Lemma 3.1** (Lemma 4.4 in [CSS19a]). *Let $\boldsymbol{R} = \boldsymbol{R}_{n,\alpha}$ for some $\alpha > 0$. For any profile $\phi \in \Phi^n$ and distribution $\boldsymbol{p} \in \Delta^{\mathcal{D}}$, there exists a pseudo distribution $\boldsymbol{q} \in \Delta_{\boldsymbol{R}}^{\mathcal{D}}$ that satisfies $\mathbb{P}(\boldsymbol{p}, \phi) \geq \mathbb{P}(\boldsymbol{q}, \phi) \geq \exp(-\alpha n - 6) \mathbb{P}(\boldsymbol{p}, \phi)$ and therefore,*

$$\max_{\boldsymbol{p} \in \Delta^{\mathcal{D}}} \mathbb{P}(\boldsymbol{p}, \phi) \geq \max_{\boldsymbol{q} \in \Delta_{\boldsymbol{R}}^{\mathcal{D}}} \mathbb{P}(\boldsymbol{q}, \phi) \geq \exp(-\alpha n - 6) \max_{\boldsymbol{p} \in \Delta^{\mathcal{D}}} \mathbb{P}(\boldsymbol{p}, \phi) \ .$$

For any probability discretization set $\mathbf{R}$, profile $\phi$ and pseudo distribution $\mathbf{q} \in \Delta_{\mathbf{R}}^{\mathcal{D}}$, define:

$$\mathbf{Z}_{\mathbf{R}}^{\phi} \stackrel{\text{def}}{=} \left\{ \mathbf{X} \in \mathbb{R}_{\geq 0}^{\ell \times [0,k]} \ \Big| \ \mathbf{X}\mathbf{1} \in \mathbb{Z}^{\ell}, [\mathbf{X}^{\top}\mathbf{1}]_j = \phi_j \text{ for all } j \in [1,k] \text{ and } \mathbf{r}^{\top}\mathbf{X}\mathbf{1} \leq 1 \right\}, \quad (3)$$

$$\mathbf{Z}_{\mathbf{R}}^{\phi,\text{frac}} \stackrel{\text{def}}{=} \left\{ \mathbf{X} \in \mathbb{R}_{\geq 0}^{\ell \times [0,k]} \ \Big| \ [\mathbf{X}^{\top}\mathbf{1}]_j = \phi_j \text{ for all } j \in [1,k] \text{ and } \mathbf{r}^{\top}\mathbf{X}\mathbf{1} \leq 1 \right\}. \quad (4)$$

The $j$'th column corresponds to frequency $m_j$ and we use $\mathbf{m}_0 \stackrel{\text{def}}{=} 0$ to capture the unseen elements. Without loss of generality, we assume $\mathbf{m}_0 < \mathbf{m}_1 < \cdots < \mathbf{m}_k$. Let $\mathbf{C}_{ij} \stackrel{\text{def}}{=} \mathbf{m}_j \log \mathbf{r}_i$ for all $i \in [\ell]$ and $j \in [0,k]$. The objective of the optimization problem is follows: for any $\mathbf{X} \in \mathbb{R}_{\geq 0}^{\ell \times [0,k]}$ define,

$$\mathbf{g}(\mathbf{X}) \stackrel{\text{def}}{=} \exp \Big( \sum_{i \in [\ell], j \in [0,k]} [\mathbf{C}_{ij}\mathbf{X}_{ij} - \mathbf{X}_{ij} \log \mathbf{X}_{ij}] + \sum_{i \in [\ell]} [\mathbf{X}\mathbf{1}]_i \log[\mathbf{X}\mathbf{1}]_i \Big) . \quad (5)$$

For any $\mathbf{q} \in \Delta_{\mathbf{R}}^{\mathcal{D}}$, the function $\mathbf{g}(\mathbf{X})$ approximates the $\mathbb{P}(\mathbf{q}, \phi)$ term and is stated below.

**Lemma 3.2** (Theorem 6.7 and Lemma 6.9 in [ACSS21]). *Let $\boldsymbol{R}$ be a probability discretization set. For any profile $\phi \in \Phi^n$ with $k$ distinct frequencies the following statements hold for $\alpha = \min(k, |\boldsymbol{R}|) \log n$: $\exp(-O(\alpha)) \cdot C_{\phi} \cdot \max_{\boldsymbol{X} \in Z_{\boldsymbol{R}}^{\phi}} \boldsymbol{g}(\boldsymbol{X}) \leq \max_{\boldsymbol{q} \in \Delta_{\boldsymbol{R}}^{\mathcal{D}}} \mathbb{P}(\boldsymbol{q}, \phi) \leq \exp(O(\alpha)) \cdot C_{\phi} \cdot \max_{\boldsymbol{X} \in Z_{\boldsymbol{R}}^{\phi}} \boldsymbol{g}(\boldsymbol{X})$ and $\max_{\boldsymbol{q} \in \Delta_{\boldsymbol{R}}^{\mathcal{D}}} \mathbb{P}(\boldsymbol{q}, \phi) \leq \exp(O(\min(k, |\boldsymbol{R}|) \log n)) \cdot C_{\phi} \cdot \max_{\boldsymbol{X} \in Z_{\boldsymbol{R}}^{\phi,\text{frac}}} \boldsymbol{g}(\boldsymbol{X})$, where $C_{\phi} \stackrel{\text{def}}{=} \frac{n!}{\prod_{j \in [1,k]} (\boldsymbol{m}_j!)^{\phi_j}}$ is a term that only depends on the profile.[2]*

The proof of concavity for the function $\mathbf{g}(\mathbf{X})$ and a running time analysis to solve the convex program are provided in [CSS19a]. For any $\mathbf{X} \in \mathbf{Z}_{\mathbf{R}}^{\phi}$, a pseudo-distributions associated with it is defined below.

**Definition 3.3.** For any $\mathbf{X} \in \mathbf{Z}_{\mathbf{R}}^{\phi}$, the discrete pseudo-distribution $\mathbf{q}_{\mathbf{X}}$ associated with $\mathbf{X}$ and $\mathbf{R}$ is defined as follows: for arbitrary $[\mathbf{X}\mathbf{1}]_i$ number of domain elements assign probability $\mathbf{r}_i$. Further $\mathbf{p}_{\mathbf{X}} \stackrel{\text{def}}{=} \mathbf{q}_{\mathbf{X}} / \|\mathbf{q}_{\mathbf{X}}\|_1$ is the distribution associated with $\mathbf{X}$ and $\mathbf{R}$.

Note that $\mathbf{q}_{\mathbf{X}}$ is a valid pseudo-distribution because of the third condition in Equation (3) and these pseudo distributions $\mathbf{p}_{\mathbf{X}}$ and $\mathbf{q}_{\mathbf{X}}$ satisfy the following lemma.

**Lemma 3.4** (Theorem 6.7 in [ACSS21]). *Let $\boldsymbol{R}$ and $\phi \in \Phi^n$ be a probability discretization set and a profile with $k$ distinct frequencies. For any $\boldsymbol{X} \in Z_{\boldsymbol{R}}^{\phi}$, the discrete pseudo distribution $\boldsymbol{q}_{\boldsymbol{X}}$ and distribution $\boldsymbol{p}_{\boldsymbol{X}}$ associated with $\boldsymbol{X}$ and $\boldsymbol{R}$ satisfy: $\exp(-O(k \log n)) C_{\phi} \cdot \boldsymbol{g}(\boldsymbol{X}) \leq \mathbb{P}(\boldsymbol{q}_{\boldsymbol{X}}, \phi) \leq \mathbb{P}(\boldsymbol{p}_{\boldsymbol{X}}, \phi)$ .*

---

[2]The theorem statement in [ACSS21] is only written with an approximation factor of $\exp(O(k \log n))$. However, their proof provides a stronger approximation factor which is upper bounded by the non-negative rank of the probability matrix, which in turn is upper bounded by the minimum of distinct frequencies and distinct probabilities. Therefore the theorem statement in [ACSS21] holds with a much stronger approximation guarantee of $\exp(O(\min(k, |\mathbf{R}|) \log n))$.

# 4 Approximate PML algorithm

Here we provide a proof sketch of Theorem 2.3 and provide a rounding algorithm that proves it. Our rounding algorithm takes as input a matrix $\mathbf{X} \in \mathbf{Z}_{\mathbf{R}}^{\phi,\mathrm{frac}}$ which may have fractional row sums and round it to integral values. This new rounded matrix $\mathbf{X}_{\mathrm{final}}$ corresponds to our approximate PML distribution (See Definition 3.3). The description of our algorithm is as follows.

---

**Algorithm 1** ApproximatePML$(\phi, \mathbf{R} = \mathbf{R}_{n,\alpha})$

---

1: Let $\mathbf{X}$ be any solution that satisfies, $\log \mathbf{g}(\mathbf{X}) \geq \max_{\mathbf{Y} \in \mathbf{Z}_{\mathbf{R}}^{\phi,\mathrm{frac}}} \log \mathbf{g}(\mathbf{Y}) - O\left(\min(k, |\mathbf{R}|) \log n\right)$.
2: $\mathbf{X}' = \mathrm{sparsify}(\mathbf{X})$.
3: $(\mathbf{A}, \mathbf{B}) = \mathrm{swapmatrixround}(\mathbf{X}')$.
4: $(\mathbf{X}_{\mathrm{final}}, \mathbf{R}_{\mathrm{final}}) = \mathrm{create}(\mathbf{A}, \mathbf{B}, \mathbf{R})$
5: Let $\mathbf{p}'$ be the distribution with respect to $\mathbf{X}_{\mathrm{final}}$ and $\mathbf{R}_{\mathrm{final}}$ (See Definition 3.3).
6: Return $\mathbf{q} = \mathrm{discretize}(\mathbf{p}', \phi, \mathbf{R})$

---

We now provide a guarantee for each of these lines of Algorithm 1. We later use these guarantees to prove our final theorem (Theorem 2.3). The guarantees of the approximate maximizer $\mathbf{X}$ computed in the first step of the algorithm are summarized in the following lemma.

**Lemma 4.1** ([CSS19a, ACSS21]). *Line 1 of the algorithm can be implemented in* $\widetilde{O}(|\mathbf{R}|k^2 + |\mathbf{R}|^2 k)$ *time and the approximate maximizer* $\mathbf{X}$ *satisfies:* $C_\phi \cdot \mathbf{g}(\mathbf{X}) \geq \exp\left(-O\left(\min(k, |\mathbf{R}|) \log n\right)\right) \max_{\mathbf{q} \in \Delta_{\mathbf{R}}^{\mathcal{D}}} \mathbb{P}(\mathbf{q}, \phi)$ .

The guarantees of the second step of our algorithm are summarized in the following lemma. Please refer to [ACSS20] for the description of the procedure $\mathrm{sparsify}$. We use this procedure so that we can assume $|\mathbf{R}| \leq k+1$ as we can ignore the zero rows of the matrix $\mathbf{X}$.

**Lemma 4.2** (Lemma 4.3 in [ACSS20]). *For any* $\mathbf{X} \in \mathbf{Z}_{\mathbf{R}}^{\phi,\mathrm{frac}}$, *the algorithm* $\mathrm{sparsify}(\mathbf{X})$ *runs in time* $\widetilde{O}(|\mathbf{R}| k^\omega)$ *and outputs* $\mathbf{X}' \in \mathbf{Z}_{\mathbf{R}}^{\phi,\mathrm{frac}}$ *such that:* $\mathbf{g}(\mathbf{X}') \geq \mathbf{g}(\mathbf{X})$ *and* $\left|\{i \in [\ell] \mid [\mathbf{X}'\overrightarrow{1}]_i > 0\}\right| \leq k+1$ .

To explain our next step, we need to define a new operation called the $\mathrm{swap}$.

**Definition 4.3.** Given a matrix $\mathbf{A}$, indices $i_1 < i_2$, $j_1 < j_2$ and a parameter $\epsilon \geq 0$, the operation $\mathrm{swap}(\mathbf{A}, i_1, i_2, j_1, j_2, \epsilon)$ outputs a matrix $\mathbf{A}'$ that satisfies,

$$\mathbf{A}'_{ij} = \begin{cases} \mathbf{A}_{i,j} + \epsilon \text{ for } i = i_1, \ j = j_1 & \mathbf{A}_{i,j} - \epsilon \text{ for } i = i_1, \ j = j_2 , \\ \mathbf{A}_{i,j} - \epsilon \text{ for } i = i_2, \ j = j_1 & \mathbf{A}_{i,j} + \epsilon \text{ for } i = i_2, \ j = j_2 , \\ \mathbf{A}_{ij} \text{ otherwise.} \end{cases} \tag{6}$$

**Definition 4.4** (Swap distance). $\mathbf{A}'$ is $x$-swap distance from $\mathbf{A}$, if $\mathbf{A}'$ can be obtained from $\mathbf{A}$ through a sequence of swap operations and the summation of the value $\epsilon$'s in these operations is at most $x$, i.e. there is a set of parameters $\{(i_1^{(s)}, i_2^{(s)}, j_1^{(s)}, j_2^{(s)}, \epsilon^{(s)})\}_{s \in [t]}$, where $\sum_{s \in [t]} \epsilon^{(s)} \leq x$, such that $\mathbf{A}^{(s)} = \mathrm{swap}(\mathbf{A}^{(s-1)}, i_1^{(s)}, i_2^{(s)}, j_1^{(s)}, j_2^{(s)}, \epsilon^{(s)})$ for $s \in [t]$, where $\mathbf{A}^{(0)} = \mathbf{A}$ and $\mathbf{A}^{(t)} = \mathbf{A}'$.

The following lemma directly follows from Definition 4.3 and Definition 4.4.

**Lemma 4.5.** *For any matrices* $A, A' \in \mathbb{R}^{s \times t}$, *if* $A'$ *is $x$-swap distance from $A$ for some $x \geq 0$, then* $A' \overrightarrow{1} = A \overrightarrow{1}$ *and* $A'^\top \overrightarrow{1} = A^\top \overrightarrow{1}$.

Recall that our objective $\mathbf{g}(\mathbf{X})$ contains two terms: (1) the linear term $\sum_{i \in [\ell], j \in [0,k]} \mathbf{C}_{ij} \mathbf{X}_{ij}$ and (2) the entropy term $\sum_{i \in [\ell]} [\mathbf{X}\overrightarrow{1}]_i \log[\mathbf{X}\overrightarrow{1}]_i - \sum_{i \in [\ell], j \in [0,k]} \mathbf{X}_{ij} \log \mathbf{X}_{ij}$. The swap operation always increases the first term, and in the following lemma we bound the loss due to the second term.

**Lemma 4.6.** *If* $A' \in \mathbb{R}^{\ell \times [0,k]}$ *is $x$-swap distance from* $A \in \mathbf{Z}_{\mathbf{R}}^{\phi,\mathrm{frac}}$, *then,* $A' \in \mathbf{Z}_{\mathbf{R}}^{\phi,\mathrm{frac}}$ *and* $\mathbf{g}(A') \geq \exp(-O(x \log n))\mathbf{g}(A)$.

One of the main contributions of our work is the following lemma, where we repeatedly apply $\mathrm{swap}$ operation to recover a matrix $\mathbf{A}$ which exhibits several nice properties as stated below.

**Lemma 4.7.** *For any matrix $A \in \mathbb{R}^{s \times t}$ ($s \leq t$) that satisfies $A^\top \vec{1} \in \mathbb{Z}_{\geq 0}^t$. The algorithm* swapmatrixround *runs in $O(s^2 t)$ time and returns matrices $A'$ and $B$ such that,*

- $A'$ *is $O(s)$-swap distance from $A$, $A' \vec{1} = A \vec{1}$ and $A'^\top \vec{1} = A^\top \vec{1}$.*

- $0 \leq B_{ij} \leq A'_{ij}$ *for all $i \in [s]$ and $j \in [t]$, $B \vec{1} \in \mathbb{Z}_{\geq 0}^s$, $B^\top \vec{1} \in \mathbb{Z}_{\geq 0}^t$ and $\|A' - B\|_1 \leq O(s)$.*

The above lemma helps us modify our matrix $\mathbf{X}$ to a new matrix $\mathbf{A}$ that we can round using the create procedure. The guarantees of this procedure are summarized below.

**Lemma 4.8** (Lemma 6.13 in [ACSS21]). *For any $A \in \mathbf{Z}_R^{\phi,\mathrm{frac}} \subseteq \mathbb{R}_{\geq 0}^{\ell \times [0,k]}$ and $B \in \mathbb{R}_{\geq 0}^{\ell \times [0,k]}$ such that $B \leq A$, $B \vec{1} \in \mathbb{Z}^\ell$, $B^\top \vec{1} \in \mathbb{Z}^{[0,k]}$ and $\|A - B\|_1 \leq t$. The algorithm* create$(A, B, R)$ *runs in time $O(\ell k)$ and returns a solution $A'$ and a probability discretization set $R'$ such that $|R'| \leq |R| + \min(k+1, t)$, $A' \in \mathbf{Z}_{R'}^\phi$ and $g(A') \geq \exp\left(-O\left(t \log n\right)\right) g(A)$.*

As our final goal is to return a distribution in $\Delta_R^{\mathcal{D}}$, we also use the following discretization lemma.

**Lemma 4.9.** *The function* discretize *takes as input a distribution $p \in \Delta^{\mathcal{D}}$ with $\ell'$ distinct probability values, a profile $\phi$, a discretization set of the form $R = R_{n,\alpha}$ for some $\alpha > 0$ and outputs a pseudo distribution $q \in \Delta_R^{\mathcal{D}}$ such that: $\mathbb{P}\left(\frac{q}{\|q\|_1}, \phi\right) \geq \exp(-O(\min(k, |R|) + \min(k, \ell') + \alpha^2 n) \log n) \mathbb{P}(p, \phi)$.*

In Section 5, we use the guarantees stated above for each line of Algorithm 1 to prove Theorem 2.3. The description of the function discretize is specified in the proof of Lemma 4.9. We describe the procedure swapmatrixround and provide a proof sketch of Lemma 4.7 in Section 4.1.

### 4.1 Description of swapmatrixround **and comparison to [ACSS20]**

Here we describe the procedure swapmatrixround and compare our rounding algorithm to [ACSS20]. Both of [ACSS20] and our approximate PML algorithm have four main lines (1-4); we differ from [ACSS20] in the key Line 3. This line in [ACSS20] invokes a procedure called matrixround that takes as input a matrix $\mathbf{A} \in \mathbb{R}^{\ell \times [0,k]}$ and outputs a matrix $\mathbf{B} \in \mathbb{R}^{\ell \times [0,k]}$ such that: $\mathbf{B} \leq \mathbf{A}$, $\mathbf{B} \vec{1} \in \mathbb{Z}_{\geq 0}^\ell$, $\mathbf{B}^\top \vec{1} \in \mathbb{Z}_{\geq 0}^{[0,k]}$ and $\|\mathbf{A} - \mathbf{B}\|_1 \leq O(\ell + k)$. Such a matrix $\mathbf{B}$ is crucial as the procedure create uses $\mathbf{B}$ to round fractional row sums of matrix $\mathbf{A}$ to integral values. The error incurred in these two steps is at most $\exp(O(\|\mathbf{A} - \mathbf{B}\|_1 \log n)) \in \exp(O((\ell + k) \log n))$. As the procedure sparsify allows us to assume $\ell \leq k+1$, we get an $\exp(-k \log n)$ approximate PML using [ACSS20]. However, the setting that we are interested in is when $\ell \ll k$; for instance when $\ell \in O(n^{1/3})$ and $k \in \Theta(\sqrt{n})$. In these settings, we desire an $\exp(-O(\min(\ell, k) \log n)) \in \exp(-O(\ell \log n))$ approximate PML. In order to get such an improved approximation using [ACSS20], we need a matrix $\mathbf{B}$ satisfying the earlier mentioned inequalities along with $\|\mathbf{A} - \mathbf{B}\|_1 \leq O(\min(k, \ell))$. However, such a matrix $\mathbf{B}$ may not exist for arbitrary matrices $\mathbf{A}$ and the best guarantee any algorithm can achieve is $\|\mathbf{A} - \mathbf{B}\|_1 \in O(\ell + k)$.

To overcome this, we introduce a new procedure called swapmatrixround that takes as input, a matrix $\mathbf{A}$ and transforms it to a new matrix $\mathbf{A}'$ that satisfies: $g(\mathbf{A}') \geq \exp(-O(\min(k, \ell) \log n)) g(\mathbf{A})$. Furthermore, this transformed matrix $\mathbf{A}'$ exhibits a matrix $\mathbf{B}$ that satisfies the guarantees: $\mathbf{B} \leq \mathbf{A}'$, $\mathbf{B} \vec{1} \in \mathbb{Z}_{\geq 0}^\ell$, $\mathbf{B}^\top \vec{1} \in \mathbb{Z}_{\geq 0}^k$ and $\|\mathbf{A}' - \mathbf{B}\|_1 \leq O(\ell)$. These matrices $\mathbf{A}'$ and $\mathbf{B}$ are nice in that we can invoke the procedure create, which would output a valid distribution with required guarantees. In the following we provide a description of the algorithm that finds these matrices $\mathbf{A}'$ and $\mathbf{B}$.

---

**Algorithm 2** swapmatrixround($\mathbf{A}$)

---

1: Let $\mathbf{A}^{(0)} = \mathbf{A}$ and $\mathbf{D}^{(0)} = 0$.
2: **for** $r = 1 \ldots \ell$ **do**
3: $\quad (\mathbf{Y}, j) = \mathrm{partialRound}(\mathbf{A}^{(r-1)}, r)$
4: $\quad \mathbf{A}^{(r)} = \mathrm{roundiRow}(\mathbf{Y}, j, r)$.
5: $\quad \mathbf{D}^{(r)} = \mathbf{D}^{(r-1)} + \mathbf{Y} - \mathbf{A}^{(r)}$.
6: **end for**
7: Return $\mathbf{A}' = \mathbf{D}^{(\ell)} + \mathbf{A}^{(\ell)}$ and $\mathbf{B} = \mathbf{A}^{(\ell)}$.

---

Our algorithm includes two main subroutines: partialRound and roundiRow. At each iteration $i$, the procedure partialRound considers row $i$ and modifies it by repeatedly applying the swap operation. This modified row is nice as the procedure roundiRow can round this row to have an integral row sum while not affecting the rows in $[i-1]$. By iterating through all rows, we get the required matrices $\mathbf{A}'$ and $\mathbf{B}$ that satisfy the required guarantees. In the remainder, we formally state the guarantees achieved by the procedures partialRound and roundiRow.

**Lemma 4.10.** *The algorithm* partialRound *takes as inputs* $X \in \mathbb{R}_{\geq 0}^{\ell \times [0,k]}$ *and* $i \in [\ell - 1]$ *that satisfies the following,* $[X\overrightarrow{1}]_{i'} \in \mathbb{Z}_{\geq 0}$ *for all* $i' \in [1, i-1]$ *and* $[X^\top \overrightarrow{1}]_j \in \mathbb{Z}_{\geq 0}$ *for all* $j \in [0, k]$, *and outputs a matrix* $Y \in \mathbb{R}_{\geq 0}^{\ell \times [0,k]}$ *and an index* $j'$ *such that:*

- *$Y$ is within $3$-swap distance from $X$.*

- *$Y_{ij'} \geq o$ and $\sum_{i'=1}^{i-1} Y_{i'j'} + Y_{ij'} - o \in \mathbb{Z}_{\geq 0}$, where $o = [X\overrightarrow{1}]_i - \lfloor [X\overrightarrow{1}]_i \rfloor$.*

*Furthermore, the running time of the algorithm is $O(\ell k)$.*

Note that by Lemma 4.5, if $Y$ is within $3$-swap distance from $X$, then $Y\overrightarrow{1} = X\overrightarrow{1}$ and $Y^\top \overrightarrow{1} = X^\top \overrightarrow{1}$.

**Lemma 4.11.** *The algorithm* roundiRow *takes as inputs* $Y \in \mathbb{R}_{\geq 0}^{\ell \times [0,k]}$, *an column index* $j \in [0, k]$ *and a row index* $i \in [\ell - 1]$ *such that:* $Y^\top \overrightarrow{1} \in \mathbb{Z}_{\geq 0}^{[0,k]}$, $Y_{ij} \geq o$ *and* $\sum_{i'=1}^{i-1} Y_{i'j} + Y_{ij} - o \in \mathbb{Z}_{\geq 0}$, *where* $o = [Y\overrightarrow{1}]_i - \lfloor [Y\overrightarrow{1}]_i \rfloor$. *Outputs a matrix* $X \in \mathbb{R}_{\geq 0}^{\ell \times [0,k]}$ *such that,*

- *$X \leq Y$ and $\|X - Y\|_1 \leq 1$.*

- *$[X\overrightarrow{1}]_{i'} = [Y\overrightarrow{1}]_{i'}$ for all $i' \in [i-1]$, $[X\overrightarrow{1}]_i \in \mathbb{Z}_{\geq 0}$, and $X^\top \overrightarrow{1} \in \mathbb{Z}_{\geq 0}^{[0,k]}$.*

We defer the description of all the missing procedures and proofs to appendix.

## 5 Proof of Main Result (Theorem 2.3)

Here we put together the results from the previous sections to prove, Theorem 2.3.

*Proof of Theorem 2.3.* Algorithm 1 achieves the guarantees of Theorem 2.3. In the remainder of the proof, we combine the guarantees of each step of the algorithm to prove the theorem. Toward this end, we first show the following two inequalities: $\mathbf{X}_{\text{final}} \in \mathbf{Z}_{\mathbf{R}_{\text{final}}}^{\phi}$ and $\mathbf{g}(\mathbf{X}_{\text{final}}) \geq \exp(-O(\min(k, |\mathbf{R}|) \log n))\mathbf{g}(\mathbf{X})$. By Lemma 4.1, the Line 1 of Algorithm 1 returns a solution $\mathbf{X} \in \mathbf{Z}_{\mathbf{R}}^{\phi,\text{frac}}$ that satisfies,

$$C_\phi \cdot \mathbf{g}(\mathbf{X}) \geq \exp\left(-O\left(\min(k, |\mathbf{R}|) \log n\right)\right) \max_{\mathbf{q} \in \Delta_{\mathbf{R}}^{\mathcal{D}}} \mathbb{P}(\mathbf{q}, \phi). \tag{7}$$

By Lemma 4.2, the Line 2 of Algorithm 1 takes input $\mathbf{X}$ and outputs $\mathbf{X}'$ such that

$$\mathbf{X}' \in \mathbf{Z}_{\mathbf{R}}^{\phi,\text{frac}} \text{ and } \mathbf{g}(\mathbf{X}') \geq \mathbf{g}(\mathbf{X}), \tag{8}$$

and $\left|\{i \in [\ell] \mid [\mathbf{X}'\overrightarrow{1}]_i > 0\}\right| \leq k + 1$. As the matrix $\mathbf{X}'$ has at most $k + 1$ non-zero rows, without loss of generality we can assume $|\mathbf{R}| \leq k + 1$ (by discarding zero rows).

As matrix $\mathbf{X}' \in \mathbf{Z}_{\mathbf{R}}^{\phi,\text{frac}}$, we have that $\mathbf{X}'$ has integral column sums and by invoking Lemma 4.7 with parameters $s = |\mathbf{R}|$ and $t = k + 1$, we get matrices $\mathbf{A}$ and $\mathbf{B}$ that satisfy guarantees of Lemma 4.7. As $[\mathbf{A}\overrightarrow{1}]_i = [\mathbf{X}'\overrightarrow{1}]_i$ for all $i \in [\ell]$, $[\mathbf{A}^\top \overrightarrow{1}]_j = [\mathbf{X}'^\top \overrightarrow{1}]_j$ for all $j \in [0, k]$ and $\mathbf{X}' \in \mathbf{Z}_{\mathbf{R}}^{\phi,\text{frac}}$, we immediately get that $\mathbf{A} \in \mathbf{Z}_{\mathbf{R}}^{\phi,\text{frac}}$. Further note that $\mathbf{A}$ is within $O(|\mathbf{R}|) = O(\min(|\mathbf{R}|, k))$-swap distance from $\mathbf{X}'$ and by Lemma 4.6 we get that $\mathbf{g}(\mathbf{A}) \geq \exp(-O(\min(|\mathbf{R}|, k) \log n))\mathbf{g}(\mathbf{X}')$. To summarize, we showed the following inequalities,

$$\mathbf{A} \in \mathbf{Z}_{\mathbf{R}}^{\phi,\text{frac}} \text{ and } \mathbf{g}(\mathbf{A}) \geq \exp(-O(\min(|\mathbf{R}|, k) \log n))\mathbf{g}(\mathbf{X}'). \tag{9}$$

Note that, Lemma 4.7 also outputs a matrix $\mathbf{B}$ that satisfies: $\mathbf{B} \leq \mathbf{A}$, $\mathbf{B}\overrightarrow{1} \in \mathbb{Z}^\ell$, $\mathbf{B}^\top \overrightarrow{1} \in \mathbb{Z}^{[0,k]}$ and $\|\mathbf{A} - \mathbf{B}\|_1 \leq O(\min(|\mathbf{R}|, k))$. These matrices $\mathbf{A}$ and $\mathbf{B}$ satisfy the conditions of Lemma 4.8 with parameter value $t = O(\min(|\mathbf{R}|, k))$. Therefore, the procedure create takes in input matrices $\mathbf{A}, \mathbf{B}$ and returns a solution $(\mathbf{X}_{\text{final}}, \mathbf{R}_{\text{final}})$ such that $|\mathbf{R}_{\text{final}}| \leq |\mathbf{R}| + \min(\mathbf{R}, k) \leq 2|\mathbf{R}|$ and,

$$\mathbf{X}_{\text{final}} \in \mathbf{Z}_{\mathbf{R}_{\text{final}}}^\phi \text{ and } \mathbf{g}(\mathbf{X}_{\text{final}}) \geq \exp(-O(\min(|\mathbf{R}|, k) \log n))\mathbf{g}(\mathbf{A}) . \tag{10}$$

As $\mathbf{X}_{\text{final}} \in \mathbf{Z}_{\mathbf{R}_{\text{final}}}^\phi$, by definition Definition 3.3 and Lemma 3.2, the distribution $\mathbf{p}'$ satisfies,

$$\mathbb{P}(\mathbf{p}', \phi) \geq \exp(-O(\min(k, |\mathbf{R}_{\text{final}}|) \log n))C_\phi \mathbf{g}(\mathbf{X}_{\text{final}}) \geq \exp(-O(\min(k, |\mathbf{R}|)) \log n))C_\phi \mathbf{g}(\mathbf{A})$$
$$\geq \exp(-O(\min(k, |\mathbf{R}|) \log n))C_\phi \mathbf{g}(\mathbf{X}') \geq \exp(-O(\min(k, |\mathbf{R}|) \log n))C_\phi \mathbf{g}(\mathbf{X})$$
$$\geq \exp(-O(\min(k, |\mathbf{R}|) \log n)) \max_{\mathbf{q} \in \Delta_\mathbf{R}^\mathcal{D}} \mathbb{P}(\mathbf{q}, \phi) .$$

In the second inequality we used Equation (10) and $|\mathbf{R}_{\text{final}}| \leq 2|\mathbf{R}|$. In the third, fourth and fifth inequalities, we used Equation (9), Equation (8) and Equation (7) respectively.

Recall we need a distribution that approximately maximizes $\max_{\mathbf{q} \in \Delta_\mathbf{R}^\mathcal{D}} \mathbb{P}(\frac{\mathbf{q}}{\|\mathbf{q}\|_1}, \phi)$ instead of just $\max_{\mathbf{q} \in \Delta_\mathbf{R}^\mathcal{D}} \mathbb{P}(\mathbf{q}, \phi)$. In the remainder of the proof we provide a procedure to output such a distribution.

For any constant $c > 0$, let $c \cdot \mathbf{R} \overset{\text{def}}{=} \{c \cdot \mathbf{r}_i \mid \mathbf{r}_i \in \mathbf{R}\}$. For any $\mathbf{q} \in \Delta_\mathbf{R}^\mathcal{D}$, as $\|\mathbf{q}\|_1$ satisfies: $\mathbf{r}_{\min} \leq \|\mathbf{q}\|_1 \leq 1$, we get that,

$$\max_{\mathbf{q} \in \Delta_\mathbf{R}^\mathcal{D}} \mathbb{P}(\frac{\mathbf{q}}{\|\mathbf{q}\|_1}, \phi) = \max_{c \in [1, 1/\mathbf{r}_{\min}]_\mathbb{R}} \max_{\mathbf{q} \in \Delta_{c \cdot \mathbf{R}}^\mathcal{D}} \mathbb{P}(\mathbf{q}, \phi) . \tag{11}$$

The above expression holds as the maximizer $\mathbf{q}^*$ of the left hand side satisfies: $\mathbf{q}^* \in \Delta_{(1/\|\mathbf{q}*\|_1) \cdot \mathbf{R}}^\mathcal{D}$. Define $C \overset{\text{def}}{=} \{(1 + \beta)^i\}_{i \in [a]}$ for some $\beta \in o(1)$, where $a \in O(\frac{1}{\beta} \log(1/\mathbf{r}_{\min}))$ is such that $\mathbf{r}_{\min}(1 + \beta)^a = 1$. For any constant $c \in [1, 1/\mathbf{r}_{\min}]_\mathbb{R}$, note that there exists a constant $c' \in C$ such that $c(1 - \beta) \leq c' \leq c$. Furthermore, for any distribution $\mathbf{q} \in \Delta_\mathbf{R}^\mathcal{D}$ with $\|\mathbf{q}\|_1 = 1/c$, note that the distribution $\mathbf{q}' = c'\mathbf{q} \in \Delta_{c' \cdot \mathbf{R}}^\mathcal{D}$ and satisfies: $\mathbb{P}(\frac{\mathbf{q}}{\|\mathbf{q}\|_1}, \phi) = \mathbb{P}(c \cdot \mathbf{q}, \phi) = \mathbb{P}(\frac{c}{c'}\mathbf{q}', \phi) = \left(\frac{c}{c'}\right)^n \mathbb{P}(\mathbf{q}', \phi)$ . Therefore we get that, $\mathbb{P}(\mathbf{q}', \phi) = \left(\frac{c'}{c}\right)^n \mathbb{P}(\frac{\mathbf{q}}{\|\mathbf{q}\|_1}, \phi) \geq (1 - \beta)^n \mathbb{P}(\frac{\mathbf{q}}{\|\mathbf{q}\|_1}, \phi) \geq \exp(-2\beta n)\mathbb{P}(\frac{\mathbf{q}}{\|\mathbf{q}\|_1}, \phi)$ . Combining this analysis with Equation (11) we get that,

$$\max_{c \in C} \max_{\mathbf{q} \in \Delta_{c \cdot \mathbf{R}}^\mathcal{D}} \mathbb{P}(\mathbf{q}, \phi) \geq \exp(-2\beta n) \max_{\mathbf{q} \in \Delta_\mathbf{R}^\mathcal{D}} \mathbb{P}(\frac{\mathbf{q}}{\|\mathbf{q}\|_1}, \phi). \tag{12}$$

For each $c > 0$ as $|\mathbf{R}| = |c \cdot \mathbf{R}|$, our algorithm (Algorithm 1) returns a distribution $\mathbf{p}_c$ that satisfies,

$$\mathbb{P}(\mathbf{p}_c, \phi) \geq \exp(-O(\min(k, |\mathbf{R}|) \log n)) \max_{\mathbf{q} \in \Delta_{c \cdot \mathbf{R}}^\mathcal{D}} \mathbb{P}(\mathbf{q}, \phi) .$$

Let $\mathbf{p}^*$ be the distribution that achieves the maximum objective value to our convex program among the distributions $\{\mathbf{p}_c\}_{c \in C}$. Then note that $\mathbf{p}^*$ satisfies: $\mathbb{P}(\mathbf{p}^*, \phi) \geq \exp(-O(\min(k, |\mathbf{R}|) \log n) - 2\beta n) \max_{\mathbf{q} \in \Delta_\mathbf{R}^\mathcal{D}} \mathbb{P}(\frac{\mathbf{q}}{\|\mathbf{q}\|_1}, \phi)$ . Substituting $\beta = \frac{\min(k, |\mathbf{R}|)}{n}$ in the previous expression, we get,

$$\mathbb{P}(\mathbf{p}^*, \phi) \geq \exp(-O(\min(k, |\mathbf{R}|) \log n)) \max_{\mathbf{q} \in \Delta_\mathbf{R}^\mathcal{D}} \mathbb{P}(\frac{\mathbf{q}}{\|\mathbf{q}\|_1}, \phi) .$$

As each of our distributions $\mathbf{p}_c$ (including $\mathbf{p}^*$) have the number of distinct probability values upper bounded by $2|\mathbf{R}|$, by applying Lemma 4.9, we get a pseudo distribution $\mathbf{q} \in \Delta_\mathbf{R}^\mathcal{D}$ with the desired guarantees. The final run time of our algorithm is $O(|C|\mathcal{T}_1) \in O(\frac{n}{\min(k, |\mathbf{R}|)} \cdot \mathcal{T}_1)$, where $\mathcal{T}_1$ is the time to implement Algorithm 1. Further note that by Lemma 3.1, without loss of generality we can assume $|\mathbf{R}| \leq n/k$. As all the lines of Algorithm 1 are polynomial in $n$, our final running time follows from the run times of each line and we conclude the proof. $\qquad \square$

## Acknowledgments and Disclosure of Funding

We would like to thank the reviewers for their valuable feedback. Researchers on this project were supported by an Amazon Research Award, a Dantzig-Lieberman Operations Research Fellowship, a Google Faculty Research Award, a Microsoft Research Faculty Fellowship, NSF CAREER Award CCF-1844855, NSF Grant CCF-1955039, a PayPal research gift, a Simons-Berkeley Research Fellowship, a Simons Investigator Award, a Sloan Research Fellowship and a Stanford Data Science Scholarship.

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
