# A  Additional definitions

**Optimal sample complexity**   The sample complexity of an estimator $\widehat{f} : \mathcal{D}^n \to \mathbb{R}$ when estimating a property $f : \Delta^{\mathcal{D}} \to \mathbb{R}$ for distributions in a collection $P \subseteq \Delta^{\mathcal{D}}$, is the number of samples $\widehat{f}$ needs to determine $f$ with accuracy $\epsilon$ and low failure probability $\delta$ for all distributions in $P$. Specifically,

$$C^{\widehat{f}}(f, P, \delta, \epsilon) \stackrel{\text{def}}{=} \min\{n \mid \mathbb{P}\left(|f(p) - \widehat{f}(x^n)| \geq \epsilon\right) \leq \delta \text{ for all } p \in P\}.$$

The sample complexity of estimating $f$ is the lowest sample complexity of any estimator,

$$C^*(f, P, \delta, \epsilon) = \min_{\widehat{f}} C^{\widehat{f}}(f, P, \delta, \epsilon).$$

In the paper, the dependency on $\delta$ is typically de-emphasized and $\delta$ is assumed to be $1/3$.

# B  Proof of Theorem 2.2

Here we provide the proof of Theorem 2.2. The proof of this statement is implicit in the analysis presented in [HS21]. Here we provide a simpler and short proof that uses the continuity lemma presented in [HS21]. For convenience, we restate this key lemma in our notation.

**Lemma B.1** (Lemma 2 in [HS21]). *Let $A \geq 2, c_0 \in (0, 1), r, s$ be arbitrary constants with $0 < s < r \leq 1/2$ and let $\mathbf{R} = \mathbf{R}_{n,r}$. Then there exists a constant $c = c(A, c_0, r, s) > 0$ such that for any distribution $\boldsymbol{p} \in \Delta^{\mathcal{D}}$, there exists a pseudo distribution $\boldsymbol{q} \in \Delta_{\mathbf{R}}^{\mathcal{D}}$ such that: for all $S \subseteq \Phi^n$, it holds that,*

$$\mathbb{P}\left(\boldsymbol{p}, S\right) \geq \mathbb{P}\left(\frac{\boldsymbol{q}}{\|\boldsymbol{q}\|_1}, S\right)^{1/(1-c_o n^{-s})} \exp(-cn^{1-2r+s})$$

$$\mathbb{P}\left(\frac{\boldsymbol{q}}{\|\boldsymbol{q}\|_1}, S\right) \geq \mathbb{P}\left(\boldsymbol{p}, S\right)^{1/(1-c_o n^{-s})} \exp(-cn^{1-2r+s})$$

*Proof of Theorem 2.2.* Let $\mathbf{p}$ be the hidden distribution. Given a profile $\phi$, let $\mathbf{q}_{\phi} \in \Delta_{\mathbf{R}}^{\mathcal{D}}$ be any pseudo distribution that satisfies,

$$\mathbb{P}\left(\frac{\mathbf{q}_{\phi}}{\|\mathbf{q}_{\phi}\|_1}, \phi\right) \geq \exp(-O(|\mathbf{R}| \log n)) \max_{\mathbf{q} \in \Delta_{\mathbf{R}}^{\mathcal{D}}} \mathbb{P}\left(\frac{\mathbf{q}}{\|\mathbf{q}\|_1}, \phi\right)$$

$$\geq \exp(-O(n^{-1/3} \log^2 n)) \max_{\mathbf{q} \in \Delta_{\mathbf{R}}^{\mathcal{D}}} \mathbb{P}\left(\frac{\mathbf{q}}{\|\mathbf{q}\|_1}, \phi\right) .$$

As $L \stackrel{\text{def}}{=} |\Delta_{\mathbf{R}}^{\mathcal{D}}| \leq \exp(O(n^{1/3} \log^2 n))$, we use $\mathbf{q}_1, \dots \mathbf{q}_L$ to denote the pseudo distributions in $\Delta_L^{\mathcal{D}}$. Let $G \stackrel{\text{def}}{=} \{\phi \in \Phi^n \mid |f(\mathbf{p}) - \widehat{f}(\phi)| \leq \epsilon\}$, that is, the set of all profiles where the estimator succeeds. Also let $S_i = \{\phi \in G \mid \mathbf{q}_{\phi} = \mathbf{q}_i\}$. Using these definitions in the remainder of the proof, we upper bound the failure probability of our estimator.

$$\mathbb{P}\left(\left|f(\frac{\mathbf{q}_{\phi}}{\|\mathbf{q}_{\phi}\|_1}) - f(\mathbf{p})\right| > \epsilon\right) \leq \mathbb{P}\left(\mathbf{p}, \Phi^n \backslash G\right) + \sum_{\{i \in [1, |L|] \mid |f(\frac{\mathbf{q}_i}{\|\mathbf{q}_i\|_1}) - f(\mathbf{p})| > \epsilon\}} \mathbb{P}\left(\mathbf{p}, S_i\right) ,$$

$$\leq \delta + \sum_{\{i \in [1, |L|] \mid |f(\frac{\mathbf{q}_i}{\|\mathbf{q}_i\|_1}) - f(\mathbf{p})| > \epsilon\}} \mathbb{P}\left(\mathbf{p}, S_i\right) .$$

In the above inequality we used that the failure probability of the estimator is at most $\delta$, that is, $\mathbb{P}\left(\mathbf{p}, \Phi^n \backslash G\right) \leq \delta$. First note that from the definitions of $\mathbf{q}_i$, $S_i$ and $\mathbf{q}_{\phi}$, we have that,

$$\mathbb{P}\left(\frac{\mathbf{q}_i}{\|\mathbf{q}_i\|_1}, S_i\right) \geq \exp(-O(n^{-1/3} \log^2 n)) \max_{\mathbf{q} \in \Delta_{\mathbf{R}}^{\mathcal{D}}} \mathbb{P}\left(\frac{\mathbf{q}}{\|\mathbf{q}\|_1}, S_i\right) .$$

Further applying Lemma B.1 with $r = 1/3$ and $s$ with any tiny constant, we get that,

$$\mathbb{P}\left(\frac{\mathbf{q}_i}{\|\mathbf{q}_i\|_1}, S_i\right) \geq \exp(-O(n^{-1/3} \log n)) \mathbb{P}\left(\mathbf{p}, S_i\right)^{1+O(n^{-c'})} \exp(-O(n^{1/3+c'})) ,$$

for any tiny constant $c' > 0$. The above expression further simplifies to,

$$\mathbb{P}\left(\frac{\mathbf{q}_i}{\|\mathbf{q}_i\|_1}, S_i\right) \geq \exp(-O(n^{1/3+c'}))\mathbb{P}\left(\mathbf{p}, S_i\right)^{1+O(n^{-c'})}.$$

Suppose $S_i$ is set such that, $\mathbb{P}\left(\mathbf{p}, S_i\right) \geq \delta^{\frac{1}{1+O(n^{-c'})}}\exp(\frac{O(n^{1/3+c'})}{1+O(n^{-c'})})$, then note that $\mathbb{P}\left(\frac{\mathbf{q}_i}{\|\mathbf{q}_i\|_1}, S_i\right) > \delta$. As the estimator provided by the conditions of the lemma succeeds on all the profiles in $S_i$, we that $|f(\mathbf{p}) - \widehat{f}(\phi)| \leq \epsilon$ for all $\phi \in S_i$. Suppose $|f(\mathbf{p}) - f(\frac{\mathbf{q}_i}{\|\mathbf{q}_i\|_1})| > 2\epsilon$, then by triangle inequality this would imply $|f(\frac{\mathbf{q}_i}{\|\mathbf{q}_i\|_1}) - \widehat{f}(\phi)| > \epsilon$ for all $\phi \in S_i$. However note that, $\mathbb{P}\left(\frac{\mathbf{q}_i}{\|\mathbf{q}_i\|_1}, S_i\right) > \delta$, and this would imply that the failure probability of the estimator is greater than $\delta$ when the underlying distribution is $\frac{\mathbf{q}_i}{\|\mathbf{q}_i\|_1}$; a contradiction. Therefore, it should be the case that $|f(\mathbf{p}) - f(\frac{\mathbf{q}_i}{\|\mathbf{q}_i\|_1})| \leq 2\epsilon$ for all $S_i$ that satisfy $\mathbb{P}\left(\mathbf{p}, S_i\right) \geq \delta^{\frac{1}{1+O(n^{-c'})}}\exp(\frac{O(n^{1/3+c'})}{1+O(n^{-c'})})$ and our failure probability is upper bounded by,

$$\mathbb{P}\left(\left|f(\frac{\mathbf{q}_\phi}{\|\mathbf{q}_\phi\|_1}) - f(\mathbf{p})\right| > \epsilon\right) \leq \delta + |L|\delta^{1-O(n^{-c'})}\exp(O(n^{1/3+c'})).$$

Further, substituting the value of $|L| \leq \exp(O(n^{1/3}\log^2 n))$, we get our desired result and we conclude our proof. □

## C  Other results

**Lemma C.1.** *For any two vectors $u, v \in \mathbb{R}_{\geq 0}^{[0,k]}$, the following inequality holds,*

$$(u^T\overrightarrow{1}\log u^T\overrightarrow{1} + v^T\overrightarrow{1}\log v^T\overrightarrow{1}) - \sum_{i\in[0,k]}(u_i\log u_i + v_i\log v_i) \leq w^T\overrightarrow{1}\log w^T\overrightarrow{1} - \sum_{i\in[0,k]}w_i\log w_i$$

*where $w = u + v$.*

*Proof.* For any $x \in \mathbb{R}_{\geq 0}^{[0,k]}$, let $f(x) \stackrel{\text{def}}{=} x^T\overrightarrow{1}\log x^T\overrightarrow{1} - \sum_{i\in[0,k]}x_i\log x_i$. Note that $f(x)$ is concave [CSS19a] and furthermore, $f(c \cdot x) = c \cdot f(x)$.

Let $w' = \frac{1}{2}u + \frac{1}{2}v$, applying concavity we get that,

$$f(w') \geq \frac{1}{2}f(u) + \frac{1}{2}f(v),$$

As $f(w) = 2f(w')$, combined with above inequality we have our proof. □

**Lemma C.2.** *For many matrices $X, Y \in \mathbb{R}^{\ell\times[0,k]}$ (where $\|X\|_1, \|Y\|_1, k, \ell \leq O(n^2)$) such that $\|X - Y\|_1 \leq \alpha$, we have that,*

$$\left|\sum_{i\in\ell, j\in[0,k]}X_{ij}\log X_{ij} - \sum_{i\in\ell, j\in[0,k]}Y_{ij}\log Y_{ij}\right| \leq O(\alpha\log n) + O(\log n).$$

*Furthermore,*

$$\left|\sum_{i\in\ell}[X\overrightarrow{1}]_i\log[X\overrightarrow{1}]_i - \sum_{i\in\ell}[Y\overrightarrow{1}]_i\log[Y\overrightarrow{1}]_i\right| \leq O(\alpha\log n) + O(\log n).$$

*Proof.* Note that $x\log x$ is $O(\log n)$-Lipschitz for $x \geq \frac{1}{n^2}$, and for any $0 \leq x_1, x_2 \leq \frac{1}{n^2}$, we have $|x_1\log x_1 - x_2\log x_2| \leq O(\frac{1}{n^2}\log n)$. As a result, for any non-negative $x_1, x_2$ that are $O(n^2)$, we have

$$|x_1\log x_1 - x_2\log x_2| \leq \left(\frac{1}{n^2} + |x_1 - x_2|\right) \cdot O(\log n)$$

Note that $\sum_{i\in\ell, j\in[0,k]} \mathbf{X}_{ij} \leq O(n^2)$ and $\sum_{i\in\ell, j\in[0,k]} \mathbf{Y}_{ij} \leq O(n^2)$. We have

$$\left| \sum_{i\in\ell, j\in[0,k]} \mathbf{X}_{ij} \log \mathbf{X}_{ij} - \sum_{i\in\ell, j\in[0,k]} \mathbf{Y}_{ij} \log \mathbf{Y}_{ij} \right| \leq \sum_{i\in\ell, j\in[0,k]} |\mathbf{X}_{ij} \log \mathbf{X}_{ij} - \mathbf{Y}_{ij} \log \mathbf{Y}_{ij}|$$

$$\leq \sum_{i\in\ell, j\in[0,k]} (\frac{1}{n^2} + |\mathbf{X}_{ij} - \mathbf{Y}_{ij}|)O(\log n) \leq (1 + \|\mathbf{X} - \mathbf{Y}\|_1) \cdot O(\log n) \leq O(\alpha \log n) + O(\log n)$$

and similarly,

$$\left| \sum_{i\in\ell} [\mathbf{X}\overrightarrow{1}]_i \log[\mathbf{X}\overrightarrow{1}]_i - \sum_{i\in\ell} [\mathbf{Y}\overrightarrow{1}]_i \log[\mathbf{Y}\overrightarrow{1}]_i \right| \leq \sum_{i\in\ell} \left| [\mathbf{X}\overrightarrow{1}]_i \log[\mathbf{X}\overrightarrow{1}]_i - [\mathbf{Y}\overrightarrow{1}]_i \log[\mathbf{Y}\overrightarrow{1}]_i \right|$$

$$\leq \sum_{i\in\ell} (\frac{1}{n^2} + |[\mathbf{X}\overrightarrow{1}]_i - [\mathbf{Y}\overrightarrow{1}]_i|)O(\log n) \leq (\frac{1}{n} + \|\mathbf{X} - \mathbf{Y}\|_1)O(\log n) \leq O(\alpha \log n) + O(\log n)$$

We conclude the proof. $\qquad\square$

## D   Description and guarantees of partialRound

Here we provide the description of partialRound and its guarantees are summarized in Lemma 4.10.

---
**Algorithm 3** partialRound$(\mathbf{X}, i)$
---
1: $\mathbf{Z} = \text{split}(\mathbf{X}, i)$.
2: $(\mathbf{W}, j) = \text{partial\_round\_special}(\mathbf{Z}, i + 1)$.
3: $\mathbf{Y} = \text{combine}(\mathbf{W}, i)$.
4: Return $(\mathbf{Y}, j)$.
---

**Lemma 4.10.** *The algorithm* partialRound *takes as inputs* $\mathbf{X} \in \mathbb{R}_{\geq 0}^{\ell\times[0,k]}$ *and* $i \in [\ell-1]$ *that satisfies the following,* $[\mathbf{X}\overrightarrow{1}]_{i'} \in \mathbb{Z}_{\geq 0}$ *for all* $i' \in [1, i-1]$ *and* $[\mathbf{X}^\top\overrightarrow{1}]_j \in \mathbb{Z}_{\geq 0}$ *for all* $j \in [0, k]$, *and outputs a matrix* $\mathbf{Y} \in \mathbb{R}_{\geq 0}^{\ell\times[0,k]}$ *and an index* $j'$ *such that:*

- *$\mathbf{Y}$ is within 3-swap distance from $\mathbf{X}$.*

- *$\mathbf{Y}_{ij'} \geq o$ and $\sum_{i'=1}^{i-1} \mathbf{Y}_{i'j'} + \mathbf{Y}_{ij'} - o \in \mathbb{Z}_{\geq 0}$, where $o = [\mathbf{X}\overrightarrow{1}]_i - \lfloor[\mathbf{X}\overrightarrow{1}]_i\rfloor$.*

*Furthermore, the running time of the algorithm is $O(\ell k)$.*

Our algorithm invokes several subroutines and in the following we provide description for each of these and also state their guarantees. The subroutine partial\_round\_special provides an algorithm that proves the Lemma 4.10 in the special case of $[\mathbf{X}\overrightarrow{1}]_i \in [0, 1)$. The description of this subroutine and its guarantees are stated below. For convenience, we first introduce some new notation and a new operation trans.

**Notation:**   For any matrix $\mathbf{X} \in \mathbb{R}_{\geq 0}^{\ell\times[0,k]}$ and indices $i_1, i_2, j_1, j_2$, we define $\mathbf{X}(i_1 : i_2, j_1 : j_2) \overset{\text{def}}{=} \sum_{i=i_1}^{i_2} \sum_{j=j_1}^{j_2} \mathbf{X}_{ij}$ and say $(i, j) \in [i_1 : i_2, j_1 : j_2]$ if $i_1 \leq i \leq i_2$ and $j_1 \leq j \leq j_2$. Therefore, $\mathbf{X}(i_1 : i_2, j_1 : j_2) = \sum_{(i,j)\in[i_1:i_2,j_1:j_2]} \mathbf{X}_{ij}$. Also note that when $i_1 \geq i_2 + 1$ or $j_1 \geq j_2 + 1$, we are summing over a empty set and $\mathbf{X}(i_1 : i_2, j_1 : j_2) \overset{\text{def}}{=} 0$.

**Lemma D.1** (Guarantees of trans). *Let $\mathbf{X} \in \mathbb{R}_{\geq 0}^{\ell\times[0,k]}$. For any indices $i_1 \leq i_2 + 1 \leq i_3 \leq i_4 + 1$, $j_1 \leq j_2 + 1 \leq j_3 \leq j_4 + 1$, and $0 \leq v \leq \min\{\mathbf{X}(i_3 : i_4, j_1 : j_2), \mathbf{X}(i_1 : i_2, j_3 : j_4)\}$, there exists $\mathbf{Y} \in \mathbb{R}_{\geq 0}^{\ell\times[0,k]}$ such that, $\mathbf{Y}$ is within $v$-swap distance from $\mathbf{X}$, and*

- *$\mathbf{Y}(i_1 : i_2, j_1 : j_2) = \mathbf{X}(i_1 : i_2, j_1 : j_2) + v$*   •  *$\mathbf{Y}(i_3 : i_4, j_3 : j_4) = \mathbf{X}(i_3 : i_4, j_3 : j_4) + v$,*
- *$\mathbf{Y}(i_1 : i_2, j_3 : j_4) = \mathbf{X}(i_1 : i_2, j_3 : j_4) - v$*   •  *$\mathbf{Y}(i_3 : i_4, j_1 : j_2) = \mathbf{X}(i_3 : i_4, j_1 : j_2) - v$.*
- *$\mathbf{Y}_{ij} = \mathbf{X}_{ij} \ \forall(i,j) \notin [i_1 : i_2, j_1 : j_2] \cup [i_1 : i_2, j_3 : j_4] \cup [i_3 : i_4, j_1 : j_2] \cup [i_3 : i_4, j_3 : j_4]$.*

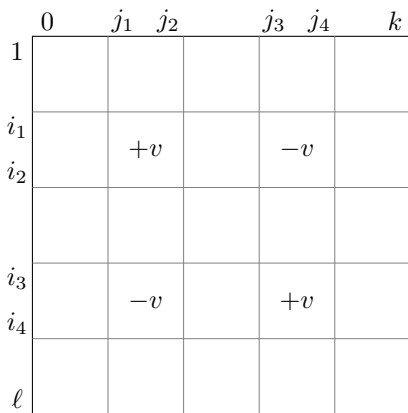

Figure 1: trans operation. It increases two blocks by $v$ respectively, and decreases two blocks by $v$ respectively.

*Furthermore, we define operation* $\mathrm{trans}(\boldsymbol{X}, v, i_1, i_2, i_3, i_4, j_1, j_2, j_3, j_4)$ *whose output is* $\boldsymbol{Y}$ *that satisfies properties above, and the running time of process* $\mathrm{trans}$ *is* $O(\ell k)$.

We defer the proof for lemma D.1 to appendix D.1. The description of our subroutine partial_round_special is as follows.

---

**Algorithm 4** partial_round_special($\mathbf{X}, i$)

---

1: $j = \min\{r : \mathbf{X}(1 : \ell, 0 : r) > \mathbf{X}(1 : i - 1, 0 : k)$.
2: $t = \mathbf{X}(1 : i - 1, 0 : k) - \mathbf{X}(1 : \ell, 0 : j - 1)$.
3: $\mathbf{Y} = \mathbf{X}$.
4: $\mathbf{Y} = \mathrm{trans}(\mathbf{Y}, \mathbf{Y}(i : i, 0 : j - 1), 1, i - 1, i, i, 0, j - 1, j, k)$.
5: **if** $\mathbf{Y}(1 : i - 1, j : j) \geq t$ **then**
6:      $v = \mathbf{Y}(1 : i - 1, j : j) - \lfloor \mathbf{Y}(1 : i - 1, j : j) \rfloor$.
7:      $\mathbf{Y} = \mathrm{trans}(\mathbf{Y}, \min\{v + 1, \mathbf{Y}(1 : i - 1, j : j) - t\}, 1, i - 1, i + 1, \ell, 0, j - 1, j, j)$.
8: **else**
9:      $v = \lceil \mathbf{Y}(1 : i - 1, j : j) \rceil - \mathbf{Y}(1 : i - 1, j : j)$.
10:      $\mathbf{Y} = \mathrm{trans}(\mathbf{Y}, v, 1, i - 1, i, \ell, j, j, j + 1, k)$.
11: **end if**
12: $\mathbf{Y} = \mathrm{trans}(\mathbf{Y}, \mathbf{Y}(i : i, j + 1 : k), i, i, i + 1, \ell, j, j, j + 1, k)$.
13: Return $(\mathbf{Y}, j)$.

---

**Lemma D.2** (Guarantee of partial_round_special). *The algorithm* partial_round_special *takes in inputs* $\boldsymbol{X} \in \mathbb{R}_{\geq 0}^{\ell \times [0,k]}$ *and* $i \in [1, \ell - 1]$ *that satisfy the conditions of Lemma 4.10 with an additions assumption that* $\sum_{j' \in [0,k]} \boldsymbol{X}_{ij'} \in [0, 1)$ *and outputs a matrix* $\boldsymbol{Y} \in \mathbb{R}_{\geq 0}^{\ell \times [0,k]}$ *and an index* $j$ *that satisfy the guarantees of Lemma 4.10. Furthermore, the running time of this procedure is* $O(\ell k)$.

We defer the proof for lemma D.2 to appendix D.2. To extend the above algorithm for the general case, we define simple operations split and combine that we define next. Intuitively, the operation combine combines $i$-th and $(i+1)$-th rows by adding them up, and split splits $i$-th row into two rows so that the summation of one row is an integer and the summation of the other row is less than one.

**Definition D.3** (Combine). For any $\mathbf{W} \in \mathbb{R}_{\geq 0}^{(\ell+1) \times [0,k]}$ and $t \in [1, \ell]$, $\mathbf{Y} \stackrel{\text{def}}{=} \mathrm{combine}(\mathbf{W}, t) \in \mathbb{R}_{\geq 0}^{\ell \times [0,k]}$ is defined as follows,

$$
\mathbf{Y}_{i,j} = \begin{cases} \mathbf{W}_{i,j} & i \leq t - 1 \\ \mathbf{W}_{i,j} + \mathbf{W}_{(i+1),j} & i = t \\ \mathbf{W}_{(i+1),j} & i \geq t + 1 \end{cases}
$$

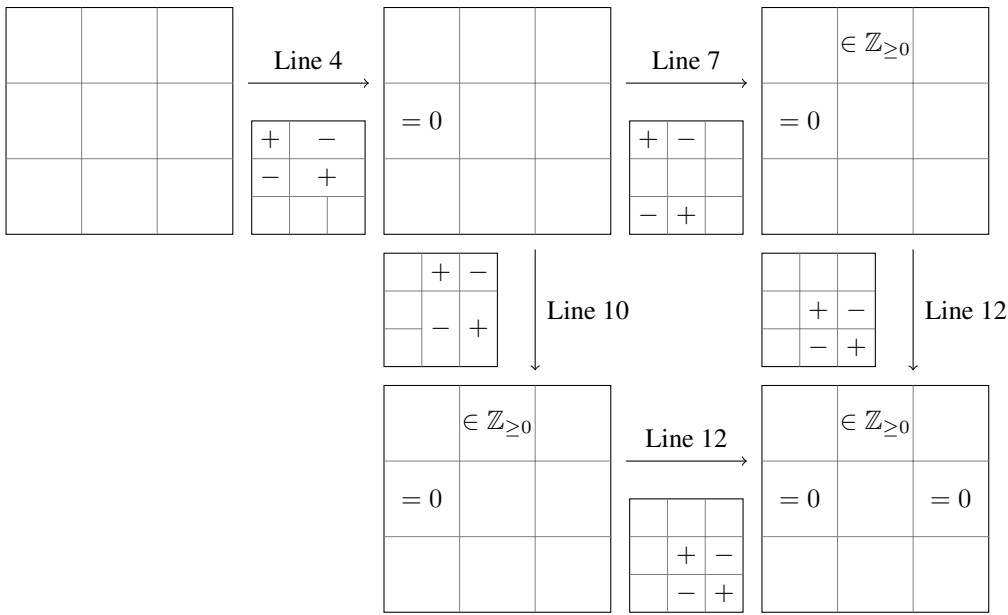

Figure 2: Algorithm 4. We use a $3 \times 3$ partitioned matrix to denote $\mathbf{Y} \in \mathbb{R}_{\geq 0}^{\ell \times [0,k]}$, where the upper (middle / lower) row denotes the first $(i-1)$ rows (the $i$-th row / the last $(\ell - i)$ rows) of $\mathbf{Y}$, and the left (middle / right) column denotes the 0-th to $(j-1)$-th columns (the $j$-th column / the last $(k-j)$ columns) of $\mathbf{Y}$. Symbol $+$ or $-$ means increasing or decreasing some elements in the block. Symbol $= 0$ means all the elements in the block are zero. Symbol $\in \mathbb{Z}_{\geq 0}$ means the sum of the block is a non-negative integer.

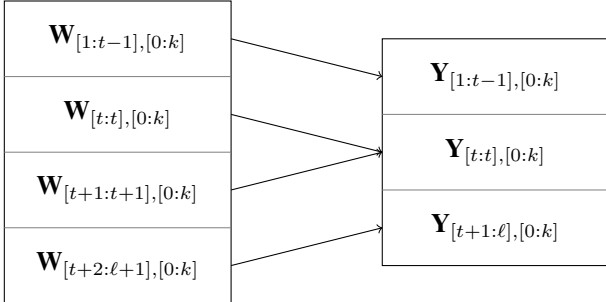

Figure 3: combine operation. It combines the $t$-th row and the $(t+1)$-th row by adding them up

**Definition D.4** (Split). Let $\mathbf{X} \in \mathbb{R}_{\geq 0}^{\ell \times [0,k]}$, $r(t) := \sum_{j \in [0,k]} \mathbf{X}_{tj}$ and $s(t) := \min\{j : \sum_{j' \leq j} \mathbf{X}_{tj'} \geq \lfloor r(t) \rfloor\}$. We define $\mathrm{split}(\mathbf{X}, t)$ to be $\mathbf{Z} \in \mathbb{R}_{\geq 0}^{(\ell+1) \times [0,k]}$ where,

$$\mathbf{Z}_{i,j} = \mathbf{X}_{i,j}, \ \ i \leq t-1 \qquad \mathbf{Z}_{i,j} = \mathbf{X}_{(i-1),j}, i \geq t+2$$

$$\mathbf{Z}_{t,j} = \begin{cases} \mathbf{X}_{t,j}, & j < s(t) \\ \lfloor r(t) \rfloor - \sum_{j' \leq j-1} \mathbf{X}_{t,j'}, & j = s(t) \\ 0, & j > s(t) \end{cases} \qquad \mathbf{Z}_{t+1,j} = \begin{cases} 0, & j < s(t) \\ \sum_{j' \leq j} \mathbf{X}_{t,j'} - \lfloor r(t) \rfloor, & j = s(t) \\ \mathbf{X}_{t,j}, & j > s(t) \end{cases}$$

We remark that $\mathbf{Z}$ has the following properties: 1) $\mathrm{combine}(\mathbf{Z}, t) = \mathbf{X}$, 2) $\sum_{j \in [0,k]} \mathbf{Z}_{tj} = \lfloor r(t) \rfloor$, 3) $\sum_{j \in [0,k]} \mathbf{Z}_{(t+1),j} = r(t) - \lfloor r(t) \rfloor$, which is a real number in $[0,1)$.

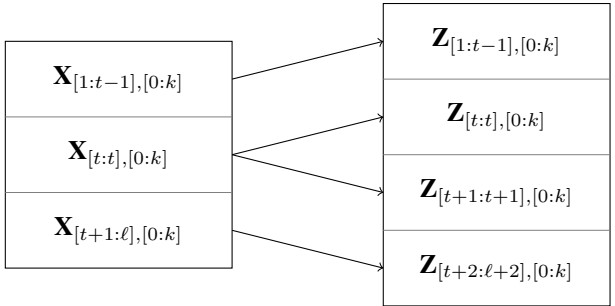

Figure 4: split operation. It splits the $t$-th row into two rows, where the sum of one row is an integer, and the sum of the other row is less than one.

### D.1 Proof for lemma D.1

*Proof for Lemma D.1.* We define $r(\mathbf{X})$ to be $\left\{(i,j) \in [i_3 : i_4, j_1 : j_2] \middle| \mathbf{X}_{ij} > 0\right\}$, and $s(\mathbf{X})$ to be $\left\{(i,j) \in [i_1 : i_2, j_3 : j_4] \middle| \mathbf{X}_{ij} > 0\right\}$. Note that $r(\mathbf{X})$ is also dependent on $i_3, i_4, j_1$ and $j_2$, but they are fixed in this proof, and $s(\mathbf{X})$ is also dependent on $i_1, i_2, j_3$ and $j_4$. We prove the lemma by induction.

When $|r(\mathbf{X})| + |s(\mathbf{X})| \leq 1$, one of $|r(\mathbf{X})|$ and $|s(\mathbf{X})|$ is 0, so $v = 0$, and $\mathbf{Y} = \mathbf{X}$ is a feasible output.

Now we assume the lemma is true when $|r(\mathbf{X})| + |s(\mathbf{X})| < t$, and show it is still true when $|r(\mathbf{X})| + |s(\mathbf{X})| = t$, where $t \geq 2$.

If $|r(\mathbf{X})| = 0$ or $|s(\mathbf{X})| = 0$, we have $v = 0$, and $\mathbf{Y} = \mathbf{X}$ is a feasible output.

If $|r(\mathbf{X})| > 0$ and $|s(\mathbf{X})| > 0$, we draw arbitrary elements $(c, b) \in r(\mathbf{X})$ and $(a, d) \in s(\mathbf{X})$.

Let $u = \min\{\mathbf{X}_{cb}, \mathbf{X}_{ad}, v\}$. Let $\mathbf{W} = \text{swap}(\mathbf{X}, a, c, b, d, u)$, and we have

- $\mathbf{W}$ is $u$-swap distance from $\mathbf{X}$, $\hspace{3cm}$ (13)
- $\mathbf{W}(i_1 : i_2, j_1 : j_2) = \mathbf{X}(i_1 : i_2, j_1 : j_2) + u$ $\quad \bullet$ $\mathbf{W}(i_3 : i_4, j_3 : j_4) = \mathbf{X}(i_3 : i_4, j_3 : j_4) + u,$ (14)
- $\mathbf{W}(i_1 : i_2, j_3 : j_4) = \mathbf{X}(i_1 : i_2, j_3 : j_4) - u$ $\quad \bullet$ $\mathbf{W}(i_3 : i_4, j_1 : j_2) = \mathbf{X}(i_3 : i_4, j_1 : j_2) - u.$ (15)
- $\forall (i,j) \notin [i_1 : i_2, j_1 : j_2] \cup [i_1 : i_2, j_3 : j_4] \cup [i_3 : i_4, j_1 : j_2] \cup [i_3 : i_4, j_3 : j_4], \mathbf{W}_{ij} = \mathbf{X}_{ij}.$ (16)

If $u = v$, $\mathbf{Y} = \mathbf{W}$ is a feasible output. Otherwise, either $u = \mathbf{X}_{cb}$ or $u = \mathbf{X}_{ad}$, so either $\mathbf{W}_{cb}$ or $\mathbf{W}_{ad}$ becomes zero. Note that $(a, b)$ and $(c, d)$ are not in $[i_1 : i_2, j_3 : j_4] \cup [i_3 : i_4, j_1 : j_2]$, so we have $|r(\mathbf{W})| + |s(\mathbf{W})| \leq |r(\mathbf{X})| + |s(\mathbf{X})| - 1$. Let $\mathbf{Y} = \text{trans}(\mathbf{W}, v - u, i_1, i_2, i_3, i_4, j_1, j_2, j_3, j_4)$, and it is a feasible output because of the following properties.

- $\mathbf{Y}$ is $(u + (v - u) = v)$-swap distance from $\mathbf{X}$,

- $\mathbf{Y}(i_1 : i_2, j_1 : j_2) = \mathbf{W}(i_1 : i_2, j_1 : j_2) + (v - u) = \mathbf{X}(i_1 : i_2, j_1 : j_2) + v,$

- $\mathbf{Y}(i_3 : i_4, j_3 : j_4) = \mathbf{W}(i_3 : i_4, j_3 : j_4) + (v - u) = \mathbf{X}(i_3 : i_4, j_3 : j_4) + v,$

- $\mathbf{Y}(i_1 : i_2, j_3 : j_4) = \mathbf{W}(i_1 : i_2, j_3 : j_4) - (v - u) = \mathbf{X}(i_1 : i_2, j_3 : j_4) - v,$

- $\mathbf{Y}(i_3 : i_4, j_1 : j_2) = \mathbf{W}(i_3 : i_4, j_1 : j_2) - (v - u) = \mathbf{X}(i_3 : i_4, j_1 : j_2) - v,$

- $\forall (i,j) \notin [i_1 : i_2, j_1 : j_2] \cup [i_1 : i_2, j_3 : j_4] \cup [i_3 : i_4, j_1 : j_2] \cup [i_3 : i_4, j_3 : j_4], \mathbf{Y}_{ij} = \mathbf{W}_{ij} = \mathbf{X}_{ij},$

where the relation between $\mathbf{Y}$ and $\mathbf{W}$ follows from inductive assumption, and the relation between $\mathbf{W}$ and $\mathbf{X}$ follows from Equation (13) to (16).

To implement trans, it is sufficient to enumerate all the elements in $r(\mathbf{X})$ and $s(\mathbf{X})$, so the running time can be $O(\ell k)$.

□

### D.2 Proof for Lemma D.2

Lemma D.2 is a special case of Lemma 4.10, with an additional input constraint $\sum_{j' \in [0,k]} \mathbf{X}_{ij'} \in [0, 1)$. In the output $\mathbf{Y}$, the $i$-th row only has one positive element $\mathbf{Y}_{ij}$. In algorithm 4, Line 4 makes $\mathbf{Y}_{ij'}$ where $j' < j$ be zero, Line 12 makes $\mathbf{Y}_{ij'}$ where $j' > j$ be zero, and Line 7 and Line 10 make $\sum_{i'=1}^{i-1} \mathbf{Y}_{i'j}$ to be an integer.

*Proof for Lemma D.2.* There are mainly two parts in this proof. The former part proves the properties of output $\mathbf{Y}$ based on Lemma D.1, and the latter part proves the parameters are valid when calling trans in Algorithm 4.

**First part of the proof** In partial_round_special$(\mathbf{X}, i)$, $\mathbf{Y}(i : i, 0 : j - 1)$ only changes at Line 4, when it decreases by $\mathbf{Y}(i : i, 0 : j - 1)$, so we have $\mathbf{Y}(i : i, 0 : j - 1) = 0$ at Line 13. $\mathbf{Y}(i : i, j + 1 : k)$ does not change after Line 12, where it decreases by $\mathbf{Y}(i : i, j + 1 : k)$, so we have $\mathbf{Y}(i : i, j + 1 : k) = 0$ at Line 13. Note that $\mathbf{Y}_{ij} = \sum_{j' \in [0,k]} \mathbf{X}_{ij'}$, so the output $\mathbf{Y}$ satisfies

$$\mathbf{Y}_{ij} = \sum_{j' \in [0,k]} \mathbf{X}_{ij'}$$

$\mathbf{Y}(1 : i - 1, j : j)$ does not change after Line 7 or Line 10. If the algorithm goes to Line 7, $\mathbf{Y}(1 : i - 1, j : j)$ decreases by $v + 1$ or $\mathbf{Y}(1 : i - 1, j : j) - t$, which means $\mathbf{Y}(1 : i - 1, j : j) = \max\left\{\left(\lfloor \mathbf{Y}(1 : i - 1, j : j) \rfloor - 1\right), t\right\}$ at Line 13. If the algorithm goes to Line 10, $\mathbf{Y}(1 : i - 1, j : j)$ increases by $v$, which means $\mathbf{Y}(1 : i - 1, j : j) = \lceil \mathbf{Y}(1 : i - 1, j : j) \rceil$ at Line 13. In both cases, $\mathbf{Y}(1 : i - 1, j : j)$ is a non-negative integer at Line 13, so the output $\mathbf{Y}$ satisfies

$$\sum_{i'=1}^{i-1} \mathbf{Y}_{i'j} \in \mathbb{Z}_{\geq 0}$$

We use three trans operations on $\mathbf{Y}$ which initially equals to $\mathbf{X}$, and we have $v \leq 1$ in each operation trans$(\mathbf{X}, v, i_1, i_2, i_3, i_4, j_1, j_2, j_3, j_4)$, so $\mathbf{Y}$ is 3-swap distance from $\mathbf{X}$.

We use $O(1)$ trans operations on $\mathbf{Y}$, so the algorithm takes $O(\ell k)$ time.

**Second part of the proof** We also need to show $0 \leq v \leq \min\{\mathbf{X}(i_3 : i_4, j_1 : j_2), \mathbf{X}(i_1 : i_2, j_3 : j_4)\}$ in each operation trans$(\mathbf{X}, v, i_1, i_2, i_3, i_4, j_1, j_2, j_3, j_4)$ to ensure the operations are valid.

Based on the definition of $j$ and $t$, we have

$$\mathbf{X}(1 : i - 1, 0 : k) = \mathbf{X}(1 : \ell, 0 : j - 1) + t,$$
$$0 \leq t < \mathbf{Y}(1 : \ell, j : j).$$

Note that $t$ and $\mathbf{Y}(1 : \ell, j : j)$ are both integers, so we have

$$t \leq \mathbf{Y}(1 : \ell, j : j) - 1.$$

Before Line 4, $\mathbf{Y} = \mathbf{X}$, and we have $\mathbf{Y}(1 : i - 1, 0 : k) \geq \mathbf{Y}(1 : \ell, 1 : j - 1)$, which implies

$$\mathbf{Y}(1 : i - 1, j : k) \geq \mathbf{Y}(i : \ell, 0 : j - 1) \geq \mathbf{Y}(i : i, 0 : j - 1),$$

which implies Line 4 is valid.

Before Line 7, we have $\mathbf{Y}(1 : i-1, 0 : k) = \mathbf{Y}(1 : \ell, 0 : j-1) + t$. Also, we have $\mathbf{Y}(i : i, 0 : j-1) = 0$ because of Line 4. So, we have

$$\mathbf{Y}(i + 1 : \ell, 0 : j - 1) \geq \mathbf{Y}(1 : i - 1, j : j) - t,$$

which implies Line 7 is valid.

Before Line 10, we have $\mathbf{Y}(1:\ell, j:j) \geq t$. Note that $t$ is an integer, so when $\mathbf{Y}(1:i-1, j:j) < t$, we have

$$\mathbf{Y}(i:\ell, j:j) = \mathbf{Y}(1:\ell, j:j) - \mathbf{Y}(1:i-1, j:j) \geq t - \mathbf{Y}(1:i-1, j:j) \geq v.$$

We also have,

$$\begin{aligned}
\mathbf{Y}(1:i-1, j+1:k) &= \mathbf{Y}(1:i-1, 0:k) - \mathbf{Y}(1:i-1, 0:j-1) - \mathbf{Y}(1:i-1, j:j) \\
&\geq (\mathbf{Y}(1:i-1, 0:k) - \mathbf{Y}(1:\ell, 0:j-1)) - \mathbf{Y}(1:i-1, j:j) \\
&= t - \mathbf{Y}(1:i-1, j:j) \geq v.
\end{aligned}$$

These two formulas imply Line 10 is valid.

After Line 7, we have either $\mathbf{Y}(1:i-1, j:j) = t$ (when $v+1 < \mathbf{Y}(1:i-1, j:j) - t$) or $\mathbf{Y}(i+1:\ell, j:j) \geq 1$ (the other case). After Line 10, we have $\mathbf{Y}(1:i-1, j:j) \leq t$. Since $t$ is an integer, when $\mathbf{Y}(1:i-1, j:j) \leq t$, we have

$$\mathbf{Y}(i:\ell, j:j) = \mathbf{Y}(1:\ell, j:j) - \mathbf{Y}(1:i-1, j:j) \geq 1.$$

As a result, we always have $\mathbf{Y}(i:\ell, j:j) \geq 1$ before Line 12. Note that $\mathbf{Y}(i:i, 0:k) < 1$ and $\mathbf{Y}(i:i, 0:j-1) = 0$, so we have

$$\mathbf{Y}(i+1:\ell, j:j) = \mathbf{Y}(i:\ell, j:j) - \mathbf{Y}_{ij} > \mathbf{Y}(i:i, 0:k) - \mathbf{Y}_{ij} = \mathbf{Y}(i:i, j+1:k),$$

which implies Line 12 is valid. $\qquad\square$

### D.3 Proof for Lemma 4.10

*Proof of Lemma 4.10.* We prove the properties mentioned in Lemma 4.10 one by one. Firstly we have

- $\mathbf{Y}_{ij} \geq \mathbf{W}_{(i+1),j} = \sum_{j' \in [0,k]} \mathbf{Z}_{(i+1),j'} = o$,

- $\sum_{i'=1}^{i-1} \mathbf{Y}_{i'j} + \mathbf{Y}_{ij} - o = \sum_{i'=1}^{i+1} \mathbf{W}_{i'j} - o = \sum_{i'=1}^{i} \mathbf{W}_{i'j} \in \mathbb{Z}_{\geq 0}$,

where the relation between $\mathbf{W}$ and $\mathbf{Z}$ is from Lemma D.2, and the relation between $\mathbf{Y}$ and $\mathbf{W}$ is from the definition of combine.

Next we show $\mathbf{Y}$ is $(3v)$-swap distance from $\mathbf{X}$. By lemma D.2, there is a set of parameters $\{(a_s, b_s, c_s, d_s, \epsilon_s)\}_{s \in [t]}$, where $\sum_{s \in [t]} \epsilon_s = 3v$, such that $A_s = \mathrm{swap}(A_{s-1}, a_s, b_s, c_s, d_s, \epsilon_s)$ for $s \in [t]$, and $A_0 = \mathbf{Z}$, $A_t = \mathbf{W}$.

Let $a_s' = a_s - \mathbb{I}(a_s > i)$ and $c_s' = c_s - \mathbb{I}(c_s > i)$ for $i \in [t]$, where $\mathbb{I}(\cdot)$ is the indicator function. Let $A_0' = \mathbf{X}$ and $A_s' = \mathrm{swap}(A_{s-1}', a_s', b_s, c_s', d_s, \epsilon_s)$ for $s \in [t]$. Then we have $A_t' = \mathbf{Y}$, which implies $\mathbf{Y}$ is $(3v)$-swap distance from $\mathbf{X}$.

We proved all the conditions of the lemma and we conclude the proof.

$\qquad\square$

## E  Description and guarantees of roundiRow

Here we provide a description of the subroutine roundiRow.

---

**Algorithm 5** roundiRow($\mathbf{Y}, j, i$)

---

1: Let matrix $\mathbf{X} \in \mathbb{R}_{\geq 0}^{\ell \times [0,k]}$.
2: Assign $\mathbf{X}_{i'j'} = \mathbf{Y}_{i'j'}$ for all $i' \in [\ell]$, $j' \in [0,k]$ and $j' \neq j$; and $\mathbf{X}_{i'j} = \mathbf{Y}_{i'j}$ for all $i' \in [1, i-1]$.

3: Assign $\mathbf{X}_{ij} = \mathbf{Y}_{ij} - o$ and residue $= 1 - o$.
4: **for** $i' = i + 1 \dots \ell$ **do**
5:     Let $w_{i'} = \min(\mathbf{Y}_{i'j}, \text{residue})$.
6:     $\mathbf{X}_{i'j} = \mathbf{Y}_{i'j} - w_{i'}$ and residue $=$ residue $- w_{i'}$.
7:     **if** residue $= 0$ **then**
8:         exit the for loop.
9:     **end if**
10: **end for**
11: **Return X**.

---

The guarantees of the above algorithm are summarized in the lemma below.

**Lemma 4.11.** *The algorithm* roundiRow *takes as inputs* $\mathbf{Y} \in \mathbb{R}_{\geq 0}^{\ell \times [0,k]}$, *an column index* $j \in [0, k]$ *and a row index* $i \in [\ell - 1]$ *such that:* $\mathbf{Y}^\top \vec{1} \in \mathbb{Z}_{\geq 0}^{[0,k]}$, $\mathbf{Y}_{ij} \geq o$ *and* $\sum_{i'=1}^{i-1} \mathbf{Y}_{i'j} + \mathbf{Y}_{ij} - o \in \mathbb{Z}_{\geq 0}$, *where* $o = [\mathbf{Y}\vec{1}]_i - \lfloor [\mathbf{Y}\vec{1}]_i \rfloor$. *Outputs a matrix* $\mathbf{X} \in \mathbb{R}_{\geq 0}^{\ell \times [0,k]}$ *such that,*

- $\mathbf{X} \leq \mathbf{Y}$ *and* $\|\mathbf{X} - \mathbf{Y}\|_1 \leq 1$.

- $[\mathbf{X}\vec{1}]_{i'} = [\mathbf{Y}\vec{1}]_{i'}$ *for all* $i' \in [i-1]$, $[\mathbf{X}\vec{1}]_i \in \mathbb{Z}_{\geq 0}$, *and* $\mathbf{X}^\top \vec{1} \in \mathbb{Z}_{\geq 0}^{[0,k]}$.

*Proof.* We first show that $\mathbf{X} \in \mathbb{R}^{\ell \times [0,k]}$ is a non-negative matrix. Note that $\mathbf{X}_{i'j'} = \mathbf{Y}_{i'j'} - o$ if $i' = i$, $j' = j$ or else $\mathbf{X}_{i'j'}$ takes one of the values in the set $\{\mathbf{Y}_{i'j'}, \mathbf{Y}_{i'j'} - \min(\mathbf{Y}_{i'j'}, \text{residue})\}$. As $\mathbf{Y}_{ij} \geq o$, we get that $\mathbf{X}_{i'j'} \geq 0$ for all $i' \in [\ell]$ and $j' \in [0, k]$. Furthermore, it is immediate that, $\mathbf{X} \leq \mathbf{Y}$.

Consider any $i' \in [1, i-1]$ and it is immediate that (Line 2) $[\mathbf{X}\vec{1}]_{i'} = [\mathbf{Y}\vec{1}]_{i'} \in \mathbb{Z}_{\geq 0}$. Furthermore, $[\mathbf{X}\vec{1}]_i = [\mathbf{Y}\vec{1}]_i - o = \lfloor [\mathbf{Y}\vec{1}]_i \rfloor \in \mathbb{Z}_{\geq 0}$.

Consider any $j' \in [0, k]$ such that $j' \neq j$ and note that $\mathbf{X}_{i'j'} = \mathbf{Y}_{i'j'}$ for all $i' \in [\ell]$. Therefore, we get that $[\mathbf{X}^\top \vec{1}]_{j'} = [\mathbf{Y}^\top \vec{1}]_{j'} \in \mathbb{Z}_{\geq 0}$. Now consider the index $j$,

$$\sum_{i' \in [\ell]} \mathbf{X}_{i'j} + 1 = \sum_{i' \in [1, i-1]} \mathbf{X}_{i'j} + \sum_{i' \in [i, \ell]} \mathbf{X}_{i'j} + 1 = \sum_{i' \in [1, i-1]} \mathbf{Y}_{i'j} + \sum_{i' \in [i, \ell]} (\mathbf{Y}_{i'j} - w_{i'}) + 1,$$

$$= \sum_{i' \in [\ell]} \mathbf{Y}_{i'j} - o - \sum_{i' \in [i+1, \ell]} w_{i'} + 1$$

The second and third equalities follow from Lines 6 and 3 of the algorithm respectively. In the following we show that, $\sum_{i' \in [i+1, \ell]} w_{i'} = 1 - o$. To show this equality all we need is to show that residue $= 0$ at the end of the loop, which holds when $\sum_{i' \in [i+1, \ell]} \mathbf{Y}_{i'j} \geq 1 - o$. As $\sum_{i' \in [\ell]} \mathbf{Y}_{i'j} \in \mathbb{Z}_{\geq 0}$, $o \in (0, 1)_{\mathbb{R}}$ and $z \overset{\text{def}}{=} \sum_{i'=1}^{i} \mathbf{Y}_{i'j} - o \in \mathbb{Z}_{\geq 0}$, we get that,

$$\sum_{i' \in [i+1, \ell]} \mathbf{Y}_{i'j} = \sum_{i' \in [\ell]} \mathbf{Y}_{i'j} - \sum_{i' \in [1, i]} \mathbf{Y}_{i'j} = \sum_{i' \in [\ell]} \mathbf{Y}_{i'j} - z - o \text{ for some integer } z \in \mathbb{Z}_{\geq 0},$$

In the following we show that $\sum_{i' \in [\ell]} \mathbf{Y}_{i'j} - z \geq 1$. As $\sum_{i' \in [\ell]} \mathbf{Y}_{i'j} - z \in \mathbb{Z}_{\geq 0}$ all we need is to show that $\sum_{i' \in [\ell]} \mathbf{Y}_{i'j} - z > 0$. Suppose $\sum_{i' \in [\ell]} \mathbf{Y}_{i'j} - z = 0$, then $\sum_{i' \in [i+1, \ell]} \mathbf{Y}_{i'j} = \sum_{i' \in [\ell]} \mathbf{Y}_{i'j} - z - o = -o < 0$, which is a contradiction as $\mathbf{Y}_{i'j'} \geq 0$ for all $i' \in [\ell]$, $j \in [0, k]$. Therefore we get that for all $j' \in [0, k]$,

$$\sum_{i' \in [\ell]} \mathbf{X}_{i'j'} + 1 = \sum_{i' \in [\ell]} \mathbf{Y}_{i'j'}. \tag{17}$$

As $\mathbf{X} \in \mathbb{R}_{\geq 0}^{\ell \times [0,k]}$ and $\sum_{i' \in [\ell]} \mathbf{Y}_{i'j} \in \mathbb{Z}_{\geq 0}$ (requirements of the lemma) we get that, $\sum_{i' \in [\ell]} \mathbf{X}_{i'j} \in \mathbb{Z}_{\geq 0}$. Therefore,

$$\mathbf{X}^\top \vec{1} \in \mathbb{Z}_{\geq 0}^{[0,k]} .$$

In the remainder of the proof we show that $\|\mathbf{X} - \mathbf{Y}\|_1 \leq 1$. Recall, earlier we showed that: $\mathbf{X} \leq \mathbf{Y}$, $[\mathbf{X}^\top \vec{1}]_{j'} = [\mathbf{Y}^\top \vec{1}]_{j'}$ for all $j' \neq j$ and $[\mathbf{X}^\top \vec{1}]_j + 1 = [\mathbf{Y}^\top \vec{1}]_j$. Combining these inequalities together, we immediately get that, $\|\mathbf{X} - \mathbf{Y}\|_1 = 1$.

We proved all the conditions of the lemma and we conclude the proof.

$\square$

## F   Guarantees of swapmatrixround

To prove Lemma 4.7, we show a stronger version of the lemma.

**Lemma F.1.** *For any matrix* $\boldsymbol{A} \in \mathbb{R}^{s \times t}$ $(s \leq t)$ *that satisfies* $\boldsymbol{A}^\top \vec{1} \in \mathbb{Z}_{\geq 0}^t$. *In algorithm* swapmatrixround, *for all* $r \in \{0\} \cup [\ell]$,

- $(\boldsymbol{D}^{(r)} + \boldsymbol{A}^{(r)})$ *is* $(3r)$-*swap distance from* $\boldsymbol{A}$.

- $\boldsymbol{D}_{ij}^{(r)} \geq 0, \boldsymbol{A}_{ij}^{(r)} \geq 0$ *for all* $i \in [s]$ *and* $j \in [t]$, $[\boldsymbol{A}^{(r)} \vec{1}]_i \in \mathbb{Z}_{\geq 0}$ *for all* $i \in [r]$, $(\boldsymbol{A}^{(r)})^\top \vec{1} \in \mathbb{Z}_{\geq 0}^t$ *and* $\|\boldsymbol{D}^{(r)}\|_1 \leq r$.

*Proof.* We prove this lemma through induction.

**Base case i=0**   In this case $\mathbf{A}^{(0)} = \mathbf{A}$, $\mathbf{D}^{(0)} = 0$, we have that all the conditions of the lemma are immediately satisfied.

**Inductive step** $r$   We assume that the lemma conditions hold for $r - 1$ and prove for the case $r$. We use $\mathbf{Y}^{(r)}$ to denote the value of $\mathbf{Y}$ in algorithm swapmatrixround when the enumerator is $r$. We verify the properties claimed by the lemma one by one, and we use the properties related to partialRound and roundiRow given by Lemma 4.10 and Lemma 4.11 respectively.

Since $(\mathbf{Y}^{(r)}, j) = \text{partialRound}(\mathbf{A}^{(r-1)}, r)$, $\mathbf{Y}^{(r)}$ is 3-swap distance from $\mathbf{A}^{(r-1)}$, which implies $(\mathbf{Y}^{(r)} + \mathbf{D}^{(r-1)})$ is 3-swap distance from $(\mathbf{A}^{(r-1)} + \mathbf{D}^{(r-1)})$. Note that $(\mathbf{A}^{(r)} + \mathbf{D}^{(r)}) = (\mathbf{Y}^{(r)} + \mathbf{D}^{(r-1)})$, and $(\mathbf{A}^{(r-1)} + \mathbf{D}^{(r-1)})$ is $(3(r-1))$-swap distance from $\mathbf{A}$, so we have $(\mathbf{D}^{(r)} + \mathbf{A}^{(r)})$ is $(3r)$-swap distance from $\mathbf{A}$.

Since $\mathbf{A}^{(r)} = \text{roundiRow}(\mathbf{Y}^{(r)}, j, r)$, we have $\mathbf{A}_{ij}^{(r)} \geq 0$ and $(\mathbf{Y}^{(r)} - \mathbf{A}^{(r)})_{ij} \geq 0$ for all $i \in [s]$ and $j \in [t]$. Furthermore, since $\mathbf{D}_{ij}^{(r-1)} \geq 0$ for all $i \in [s]$ and $j \in [t]$, and $\mathbf{D}^{(r)} = (\mathbf{Y}^{(r)} - \mathbf{A}^{(r)}) + \mathbf{D}^{(r-1)}$, we have $\mathbf{D}_{ij}^{(r)} \geq 0$ for all $i \in [s]$ and $j \in [t]$.

Since $(\mathbf{Y}^{(r)}, j) = \text{partialRound}(\mathbf{A}^{(r-1)}, r)$, we have $[\mathbf{Y}^{(r)} \vec{1}]_i = [\mathbf{A}^{(r-1)} \vec{1}]_i \in \mathbb{Z}_{\geq 0}$ for $i \in [r-1]$. Since $\mathbf{A}^{(r)} = \text{roundiRow}(\mathbf{Y}^{(r)}, j, r)$, we have $[\mathbf{A}^{(r)} \vec{1}]_i = [\mathbf{Y}^{(r)} \vec{1}]_i \in \mathbb{Z}_{\geq 0}$ for $i \in [r-1]$, and $[\mathbf{A}^{(r)} \vec{1}]_i \in \mathbb{Z}_{\geq 0}$ for $i = r$. As a result, $[\mathbf{A}^{(r)} \vec{1}]_i \in \mathbb{Z}_{\geq 0}$ for $i \in [r]$.

Since $\mathbf{A}^{(r)} = \text{roundiRow}(\mathbf{Y}^{(r)}, j, r)$, we have $(\mathbf{A}^{(r)})^\top \vec{1} \in \mathbb{Z}_{\geq 0}^t$.

Since $\mathbf{A}^{(r)} = \text{roundiRow}(\mathbf{Y}^{(r)}, j, r)$, we have $\|\mathbf{Y}^{(r)} - \mathbf{A}^{(r)}\|_1 \leq 1$. Then we have $\|\mathbf{D}^{(r)}\|_1 \leq \|\mathbf{Y}^{(r)} - \mathbf{A}^{(r)}\|_1 + \|\mathbf{D}^{(r-1)}\|_1 \leq r$ because $\mathbf{D}^{(r)} = (\mathbf{Y}^{(r)} - \mathbf{A}^{(r)}) + \mathbf{D}^{(r-1)}$.

$\square$

**Corollary F.2.** *For any matrix* $\boldsymbol{A} \in \mathbb{R}^{s \times t}$ $(s \leq t)$ *that satisfies* $\boldsymbol{A}^\top \vec{1} \in \mathbb{Z}_{\geq 0}^t$. *The algorithm* swapmatrixround *returns matrices* $\boldsymbol{A}'$ *and* $\boldsymbol{B}$ *such that,*

- $\boldsymbol{A}'$ *is* $O(s)$-*swap distance from* $\boldsymbol{A}$.

- $0 \leq \boldsymbol{B}_{ij} \leq \boldsymbol{A}'_{ij}$ for all $i \in [s]$ and $j \in [t]$, $\boldsymbol{B}\overrightarrow{1} \in \mathbb{Z}^s_{\geq 0}$, $\boldsymbol{B}^\top \overrightarrow{1} \in \mathbb{Z}^t_{\geq 0}$ and $\|\boldsymbol{A}' - \boldsymbol{B}\|_1 \leq O(s)$.

$\mathbf{A}'$ is $O(s)$-swap distance from $\mathbf{A}$ , and it implies $\mathbf{A}' \overrightarrow{1} = \mathbf{A} \overrightarrow{1}$ and $\mathbf{A}'^\top \overrightarrow{1} = \mathbf{A}^\top \overrightarrow{1}$.

We run $O(s)$ times $\mathrm{partialRound}$, $\mathrm{roundiRow}$, and matrix addition and subtraction, so the running time is $O(s^2 t)$.

We proved all the conditions of Lemma 4.7 and we conclude the proof.

# G  Proof for Lemma 4.9

*Proof.* Let $\{\mathbf{r}'_i\}_{i \in [\ell']}$ be the set of distinct probability values of distribution $\mathbf{p}$. Let $\mathbf{X} \in \mathbb{R}^{\ell' \times [0,k]}_{\geq 0}$ be the maximizer of the following optimization problem,

$$\max_{\mathbf{Y} \in \mathbf{Z}^\phi_\mathbf{p}} \mathbf{g}(\mathbf{Y}) \ .$$

By Lemma 3.2, the maximizer $\mathbf{X}$ satisfies,

$$C_\phi \cdot \mathbf{g}(\mathbf{X}) \geq \exp(-O(\min(k, \ell'))\log n)\mathbb{P}(\mathbf{p}, \phi) \ ,$$

In the following we consider multiple distributions with different probabilities to denote the probability values explicitly, we extend the notation of $\mathbf{g}(\mathbf{X})$ to $\mathbf{g}(\mathbf{X}, \mathbf{r})$. Given the solution $\mathbf{X}$, we now maximize over the probability values, that is, solve,

$$\max_{\mathbf{r}'' \in [0,1]^{\ell'}} \mathbf{g}(\mathbf{X}, \mathbf{r}'') \overset{\mathrm{def}}{=} \sum_{i \in [\ell'], j \in [0,k]} (\mathbf{m}_j \log \mathbf{r}''_i \mathbf{X}_{ij} - \mathbf{X}_{ij} \log \mathbf{X}_{ij}) + \sum_{i \in [\ell']} [\mathbf{X}\mathbf{1}]_i \log[\mathbf{X}\mathbf{1}]_i \ ,$$

$$\text{such that } \sum_{i \in [\ell']} [\mathbf{X}\mathbf{1}]_i \mathbf{r}''_i = 1 \ .$$

The optimum solution $\mathbf{r}^*$ satisfies $\sum_{i \in [\ell']} [\mathbf{X}\mathbf{1}]_i \mathbf{r}^*_i = 1$ and,

$$\mathbf{r}^*_i = \frac{\sum_{j \in [0,k]} \mathbf{m}_j \mathbf{X}_{ij}}{[\mathbf{X}\overrightarrow{1}]_i \sum_{i' \in [\ell']} \sum_{j \in [0,k]} \mathbf{m}_j \mathbf{X}_{i'j}} = \frac{\sum_{j \in [0,k]} \mathbf{m}_j \mathbf{X}_{ij}}{n[\mathbf{X}\overrightarrow{1}]_i} \ .$$

Furthermore,

$$C_\phi \mathbf{g}(\mathbf{X}, \mathbf{r}^*) \geq C_\phi \mathbf{g}(\mathbf{X}, \mathbf{r}') \geq \exp(-O(\min(k, \ell'))\log n)\mathbb{P}(\mathbf{p}, \phi) \ . \tag{18}$$

Substituting values of $\mathbf{r}^*$ in $\mathbf{g}(\mathbf{X}, \mathbf{r}^*)$ we get,

$$\mathbf{g}(\mathbf{X}, \mathbf{r}^*) = n \sum_{i \in [\ell']} [\mathbf{X}\overrightarrow{1}]_i \mathbf{r}^*_i \log \mathbf{r}^*_i - \sum_{i \in [\ell'], j \in [0,k]} \mathbf{X}_{ij} \log \mathbf{X}_{ij} + \sum_{i \in [\ell']} [\mathbf{X}\mathbf{1}]_i \log[\mathbf{X}\mathbf{1}]_i \tag{19}$$

We now construct a pseudo distribution $\mathbf{q}$ by rounding down the probability values into set $\mathbf{R}$ as follows: for each $i \in [\ell]$, the number of elements with probability value $\mathbf{r}_i$ is equal to $\sum_{\{i' \in [\ell'] \ | \ \lfloor \mathbf{r}^*_{i'} \rfloor_\mathbf{R} = \mathbf{r}_i\}} [\mathbf{X}\overrightarrow{1}]_{i'}$, where $\lfloor y \rfloor_\mathbf{R} \overset{\mathrm{def}}{=} \max_{\{x \in \mathbf{R} \ | \ x \leq y\}} x$. Define a new solution $\mathbf{X}' \in \mathbb{R}^{[\ell] \times [0,k]}$ as follows: $\mathbf{X}'_{ij} \overset{\mathrm{def}}{=} \sum_{\{i' \in [\ell'] \ | \ \lfloor \mathbf{r}^*_{i'} \rfloor_\mathbf{R} = \mathbf{r}_i\}} \mathbf{X}_{i'j}$. Let $\beta = \sum_{i' \in [\ell']} \lfloor \mathbf{r}^*_{i'} \rfloor_\mathbf{R} [\mathbf{X}\overrightarrow{1}]_{i'} = \sum_{i \in [\ell]} \mathbf{r}_i [\mathbf{X}'\overrightarrow{1}]_i$ and note that $\|\mathbf{q}\|_1 = \beta$. Now note that the following inequality immediately holds,

$$\mathbb{P}\left(\frac{\mathbf{q}}{\|\mathbf{q}\|_1}, \phi\right) = \mathbb{P}(\mathbf{q}, \phi)\beta^{-n} \text{ and } \mathbb{P}(\mathbf{q}, \phi) \geq C_\phi \mathbf{g}(\mathbf{X}', \mathbf{r}). \tag{20}$$

Now consider,

$$\log \mathbf{g}(\mathbf{X}', \mathbf{r}) = \sum_{i \in [\ell], j \in [0,k]} (\mathbf{m}_j \log \mathbf{r}_i \mathbf{X}'_{ij} - \mathbf{X}'_{ij} \log \mathbf{X}'_{ij}) + \sum_{i \in [\ell]} [\mathbf{X}'\mathbf{1}]_i \log[\mathbf{X}'\mathbf{1}]_i,$$

$$\geq \sum_{i \in [\ell'], j \in [0,k]} \mathbf{m}_j \log \lfloor \mathbf{r}^*_{i'} \rfloor_\mathbf{R} \mathbf{X}_{ij} - \sum_{i \in [\ell'], j \in [0,k]} \mathbf{X}_{ij} \log \mathbf{X}_{ij} + \sum_{i \in [\ell']} [\mathbf{X}\mathbf{1}]_i \log[\mathbf{X}\mathbf{1}]_i \ . \tag{21}$$

Consider the first term in the above expression,

$$\sum_{i\in[\ell'],j\in[0,k]} \mathbf{m}_j \log\lfloor \mathbf{r}_{i'}^* \rfloor_{\mathbf{R}} \mathbf{X}_{ij} = \sum_{i\in[\ell'],j\in[0,k]} \mathbf{m}_j \log \frac{\lfloor \mathbf{r}_i^* \rfloor_{\mathbf{R}}}{\beta} \mathbf{X}_{ij} + \log\beta \sum_{i\in[\ell'],j\in[0,k]} \mathbf{m}_j \mathbf{X}_{ij},$$

$$= \sum_{i\in[\ell'],j\in[0,k]} \mathbf{m}_j \log \frac{\lfloor \mathbf{r}_i^* \rfloor_{\mathbf{R}}}{\beta} \mathbf{X}_{ij} + n\log\beta \tag{22}$$

To simplify the above expression we define $\alpha_i$ as, $\mathbf{r}_i^*(1+\alpha_i) = \frac{\lfloor \mathbf{r}_i^* \rfloor_{\mathbf{R}}}{\beta}$, then we get that,

$$\sum_{i\in[\ell']} \mathbf{r}_i^*(1+\alpha_i)[\mathbf{X}\overrightarrow{1}]_i = \sum_{i\in[\ell']} \frac{\lfloor \mathbf{r}_i^* \rfloor_{\mathbf{R}}}{\beta}[\mathbf{X}\overrightarrow{1}]_i = 1.$$

However, we also have that, $\sum_{i\in[\ell']} \mathbf{r}_i^*[\mathbf{X}\overrightarrow{1}]_i = 1$. Combining both we get that, $\sum_{i\in[\ell']} \alpha_i\mathbf{r}_i^*[\mathbf{X}\overrightarrow{1}]_i = 0$. Substituting this equality in Equation 22, we get,

$$\sum_{i\in[\ell'],j\in[0,k]} \mathbf{m}_j \log\lfloor \mathbf{r}_{i'}^* \rfloor_{\mathbf{R}} \mathbf{X}_{ij} = \sum_{i\in[\ell'],j\in[0,k]} \mathbf{m}_j \log \frac{\lfloor \mathbf{r}_i^* \rfloor_{\mathbf{R}}}{\beta} \mathbf{X}_{ij} + n\log\beta$$

$$= \sum_{i\in[\ell'],j\in[0,k]} \mathbf{m}_j \log \mathbf{r}_i^*(1+\alpha_i)\mathbf{X}_{ij} + n\log\beta$$

$$= \sum_{i\in[\ell'],j\in[0,k]} \mathbf{m}_j \log \mathbf{r}_i^*\mathbf{X}_{ij} + \sum_{i\in[\ell'],j\in[0,k]} \mathbf{m}_j \log(1+\alpha_i)\mathbf{X}_{ij} + n\log\beta$$

$$\geq \sum_{i\in[\ell'],j\in[0,k]} \mathbf{m}_j \log \mathbf{r}_i^*\mathbf{X}_{ij} + \sum_{i\in[\ell'],j\in[0,k]} \mathbf{m}_j(\alpha_i - \alpha_i^2)\mathbf{X}_{ij} + n\log\beta$$

$$\geq \sum_{i\in[\ell'],j\in[0,k]} \mathbf{m}_j \log \mathbf{r}_i^*\mathbf{X}_{ij} - \sum_{i\in[\ell'],j\in[0,k]} \mathbf{m}_j\alpha_i^2\mathbf{X}_{ij} + n\log\beta . \tag{23}$$

In the fourth inequality we used the inequality $\log(1+x) \geq x - x^2$, when $x \in (0,1)$. In the last inequality we used $\sum_{i\in[\ell']} \alpha_i\mathbf{r}_i^*[\mathbf{X}\overrightarrow{1}]_i = 0$. Now note that each of the $\alpha_i$ satisfy, $|\alpha_i| \leq O(\alpha)$. Substituting it in the above expression we get,

$$\sum_{i\in[\ell'],j\in[0,k]} \mathbf{m}_j \log\lfloor \mathbf{r}_{i'}^* \rfloor_{\mathbf{R}} \mathbf{X}_{ij} \geq \sum_{i\in[\ell'],j\in[0,k]} \mathbf{m}_j \log \mathbf{r}_i^*\mathbf{X}_{ij} - \alpha^2 n + n\log\beta, \tag{24}$$

where in the above inequality we used $\sum_{i\in[\ell'],j\in[0,k]} \mathbf{m}_j\mathbf{X}_{ij} = \sum_{j\in[0,k]} \mathbf{m}_j\phi_j = n$. Combining the above inequality with Equation (21), we get,

$$\log\mathbf{g}(\mathbf{X}',\mathbf{r}) \geq \sum_{i\in[\ell'],j\in[0,k]} \mathbf{m}_j \log\lfloor \mathbf{r}_{i'}^* \rfloor_{\mathbf{R}} \mathbf{X}_{ij} - \sum_{i\in[\ell'],j\in[0,k]} \mathbf{X}_{ij}\log\mathbf{X}_{ij} + \sum_{i\in[\ell']} [\mathbf{X1}]_i \log[\mathbf{X1}]_i,$$

$$\geq \sum_{i\in[\ell'],j\in[0,k]} \mathbf{m}_j \log \mathbf{r}_i^*\mathbf{X}_{ij} - \sum_{i\in[\ell'],j\in[0,k]} \mathbf{X}_{ij}\log\mathbf{X}_{ij} + \sum_{i\in[\ell']} [\mathbf{X1}]_i \log[\mathbf{X1}]_i - \alpha^2 n + n\log\beta,$$

$$= \log\mathbf{g}(\mathbf{X},\mathbf{r}^*) - \alpha^2 n + n\log\beta \tag{25}$$

Combining everything we get the following,

$$\log\mathbb{P}(\mathbf{p},\phi) - \log\mathbb{P}\left(\frac{\mathbf{q}}{\|\mathbf{q}\|_1},\phi\right) = \log\mathbb{P}(\mathbf{p},\phi) - \log\mathbb{P}(\mathbf{q},\phi) + n\log\beta,$$

$$\leq O(\min(k,\ell')\log n) + \log(C_\phi\mathbf{g}(\mathbf{X},\mathbf{r}^*)) - \log(C_\phi\mathbf{g}(\mathbf{X}',\mathbf{r})) + n\log\beta,$$

$$\leq O(\min(k,\ell')\log n) + \alpha^2 n .$$

In the first, second and third inequalities we use Equation (18), Equation (20) and Equation (25) respectively. We conclude the proof. $\square$

# H  Proof for Lemma 4.6

We firstly show a special case of Lemma 4.6

**Lemma H.1.** *If $A' = \mathrm{swap}(A, i_1, i_2, j_1, j_2, \epsilon)$ where $A, A' \in \mathbb{R}^{\ell \times [0,k]}$ and $A \in \mathbf{Z}_{\mathbf{R}}^{\phi,\mathrm{frac}}$, then, $A' \in \mathbf{Z}_{\mathbf{R}}^{\phi,\mathrm{frac}}$ and $\mathbf{g}(A') \geq \exp(-4\epsilon \log n) \, \mathbf{g}(A)$.*

*Proof.* Since $\mathbf{A}' = \mathrm{swap}(\mathbf{A}, i_1, i_2, j_1, j_2, \epsilon)$, we have $\mathbf{A}'^\top \overrightarrow{1} = \mathbf{A}^\top \overrightarrow{1}$ and $\mathbf{A}' \overrightarrow{1} = \mathbf{A} \overrightarrow{1}$. As $\mathbf{A} \in \mathbf{Z}_{\mathbf{R}}^{\phi,\mathrm{frac}}$, we immediately get that $\mathbf{A}' \in \mathbf{Z}_{\mathbf{R}}^{\phi,\mathrm{frac}}$ and we also have

$$
\frac{\mathbf{g}(\mathbf{A}')}{\mathbf{g}(\mathbf{A})} = \frac{\exp\left( \sum_{i\in[\ell], j\in[0,k]} \left[ \mathbf{C}_{ij}\mathbf{A}'_{ij} - \mathbf{A}'_{ij}\log\mathbf{A}'_{ij} \right] + \sum_{i\in[\ell]}[\mathbf{A}'\mathbf{1}]_i \log[\mathbf{A}'\mathbf{1}]_i \right)}{\exp\left( \sum_{i\in[\ell], j\in[0,k]} \left[ \mathbf{C}_{ij}\mathbf{A}_{ij} - \mathbf{A}_{ij}\log\mathbf{A}_{ij} \right] + \sum_{i\in[\ell]}[\mathbf{A}\mathbf{1}]_i \log[\mathbf{A}\mathbf{1}]_i \right)}
$$

$$
= \frac{\exp\left( \sum_{i\in[\ell], j\in[0,k]} \left[ \mathbf{C}_{ij}\mathbf{A}'_{ij} - \mathbf{A}'_{ij}\log\mathbf{A}'_{ij} \right] \right)}{\exp\left( \sum_{i\in[\ell], j\in[0,k]} \left[ \mathbf{C}_{ij}\mathbf{A}_{ij} - \mathbf{A}_{ij}\log\mathbf{A}_{ij} \right] \right)}
$$

$$
\geq \frac{\exp\left( (\mathbf{C}_{i_1 j_1} - \mathbf{C}_{i_1 j_2} - \mathbf{C}_{i_2 j_1} + \mathbf{C}_{i_2 j_2})\epsilon \right)}{\exp\left( \mathbf{A}'_{i_1 j_1}\log\mathbf{A}'_{i_1 j_1} - \mathbf{A}_{i_1 j_1}\log\mathbf{A}'_{i_1 j_1} + \mathbf{A}'_{i_2 j_2}\log\mathbf{A}'_{i_2 j_2} - \mathbf{A}_{i_2 j_2}\log\mathbf{A}'_{i_2 j_2} \right)}
$$

$$
\geq \frac{1}{\exp\left( 2\epsilon\log(n+1) \right)} \geq \exp\left( -4\epsilon\log n \right),
$$

where the last inequality holds because because $\frac{d}{dx}(x\log(x)) = (\log(x) + 1)$ and

$$
\mathbf{C}_{i_1 j_1} - \mathbf{C}_{i_1 j_2} - \mathbf{C}_{i_2 j_1} + \mathbf{C}_{i_2 j_2} = \mathbf{m}_{j_1}\mathbf{r}_{i_1} - \mathbf{m}_{j_2}\mathbf{r}_{i_1} - \mathbf{m}_{j_1}\mathbf{r}_{i_2} + \mathbf{m}_{j_2}\mathbf{r}_{i_2}
$$
$$
= (\mathbf{m}_{j_2} - \mathbf{m}_{j_1})(\mathbf{r}_{i_2} - \mathbf{r}_{i_1}) \geq 0.
$$

$\square$

*Proof for Lemma 4.6.* Since $\mathbf{A}'$ is $x$-swap distance from $\mathbf{A}$, there exists a set of parameters denoted by $\{(i_1^{(s)}, i_2^{(s)}, j_1^{(s)}, j_2^{(s)}, \epsilon^{(s)})\}_{s\in[t]}$, where $\sum_{s\in[t]} \epsilon^{(s)} \leq x$, s.t $\mathbf{A}^{(s)} = \mathrm{swap}(\mathbf{A}^{(s-1)}, i_1^{(s)}, i_2^{(s)}, j_1^{(s)}, j_2^{(s)}, \epsilon^{(s)})$ for $s \in [t]$, where $\mathbf{A}^{(0)} = \mathbf{A}$, $\mathbf{A}^{(t)} = \mathbf{A}'$.

We apply Lemma H.1 on $\mathbf{A}^{(i)}$ and $\mathbf{A}^{(i-1)}$ for each $i \in [t]$, and we have

$$
\mathbf{A}^{(0)} \in \mathbf{Z}_{\mathbf{R}}^{\phi,\mathrm{frac}} \Rightarrow \mathbf{A}^{(1)} \in \mathbf{Z}_{\mathbf{R}}^{\phi,\mathrm{frac}} \Rightarrow \mathbf{A}^{(2)} \in \mathbf{Z}_{\mathbf{R}}^{\phi,\mathrm{frac}} \Rightarrow \cdots \Rightarrow \mathbf{A}^{(t)} \in \mathbf{Z}_{\mathbf{R}}^{\phi,\mathrm{frac}},
$$

$$
\mathbf{g}(\mathbf{A}^{(t)}) \geq \exp\left( -4\epsilon^{(t)}\log n \right) \mathbf{g}(\mathbf{A}^{(t-1)}) \geq \cdots \geq \exp\left( -4\left( \sum_{s\in[t]} \epsilon^{(s)} \right)\log n \right) \mathbf{g}(\mathbf{A}^{(0)}),
$$

which implies $\mathbf{A}' \in \mathbf{Z}_{\mathbf{R}}^{\phi,\mathrm{frac}}$ and $\mathbf{g}(\mathbf{A}') \geq \exp\left( -O\left( x\log n \right) \right) \mathbf{g}(\mathbf{A})$. We conclude the proof.  $\square$