# OpenReview forum: "On the Efficient Implementation of High Accuracy Optimality of Profile Maximum Likelihood"
_NeurIPS.cc/2022/Conference — NeurIPS 2022 Accept_

### Official Review · Reviewer_w8eP · 2022-07-15

**Rating:** 5
**Confidence:** 2
**Soundness:** 2 fair
**Presentation:** 1 poor
**Contribution:** 3 good

**Summary:**

The submission considers the problem of estimating some symmetric properties of finite-supported discrete distributions. The paper is interested in a very specific type of such methods, an "approximate profile maximum likelihood (PML) estimator", a variant of MLE optimized on a more compactly defined objectives. The paper concerns a highly technical question: can any efficient and approximate PML estimator be and sample competitive in the regime $\epsilon \gg n^{-1/3}$? Maybe I should summarize a few preliminaries as well:

- "Sample-competitive" of an estimator means: the failure probability of the estimator matches the standard $\exp(-\Omega(n\epsilon^2))$ failure probability.

- The exact PML estimator is shown to be sample-optimal in $\epsilon \gg n^{-1/3}$ regime, and cannot be sample-optimal below the $n^{-1/3}$ threshold.

- Existing *approximate* PML estimators are shown to be sample optimal in the regime $\epsilon \gg n^{-1/4}$. Can they tighten this to $n^{-1/3}$?

They answers this question positively, closing the problem (in my guess). For this, the paper improves the algorithm proposed in [ACSS20, ACSS21]. Main technical innovation seems to be a certain swap operator, which is hard for me to follow further.

**Questions:**

- Can you give some comments on why do we care an approximate PML estimator in the intro explicitly?

- It would have been helpful to make more contrast between this paper and previous work in [ACSS21]. e.g., why previous work can only work in $\epsilon \gg n^{-1/4}$.

**Limitations:**

I do not see any negative societal impact.

**Strengths And Weaknesses:**

It seems like that the paper is mostly an extension of [ACSS21], but I feel that the paper is not sufficiently self-contained. Overall the presentation is very hard to follow. It might be just because I am not familiar with the problem. But even when considering that, in my opinion, the authors should have more taken care of broader audiences, given that the scope of the paper is very narrow and specific. For instance, Section 3 and 4 are extremely technical and very hard to follow. Most of the results (theorems and lemmas) seem to be coming from [ACSS21] without much explanation. I could not evaluate the merit of the paper further if not I just believe their claim.

---

> ### Author Response · Authors · 2022-08-02
> **Response to questions and writing concerns raised by the reviewer**
>
> Thank you for your time and valuable feedback. We appreciate your writing suggestions. We respond to specific points below.
>
> {\bf Concerns related to writing}: Thank you for the feedback, in the final revision, we will add more background in the introduction and other areas to make the paper self-contained and also accessible to a broader audience. The technical nature of Sections 3 and 4 was due in part to (i) space constraints and (ii) the prioritization of highlighting our new contributions. In the final submission, we will seek to add additional intuitive explanations for lemmas, connections between related results, and highlights of our technical contributions in comparison to prior work.
>
> {\bf Comparison of our results to [ACSS21] and the merits of our paper}: Please refer to the results section starting at Line 147 and also in Section 4.1 starting at Line 265 for a comparison of our work to the previous approximate PML algorithms including [ACSS21].  In the revision, we will add a version of the following explanations to help compare our results with [ACSS21] and better evaluate the merits of our paper.
>
> Prior approximate PML algorithms, e.g. [CSS19, ACSS20, ACSS21], have two main steps: 1) solve the convex approximation of the PML. 2) round the fractional solution returned by the previous step. Both these steps incur a loss in the approximation ratio. One of the key differences between many approximate PML algorithms, e.g. [CSS19, ACSS20, ACSS21] and our own algorithm is in the second step (rounding algorithm). Both the previous best [ACSS20] and our algorithm, use the same convex approximation from [CSS19, ACSS21] but provide different rounding algorithms with improved theoretical guarantees. The main contribution and key difference from the previous best [ACSS20] algorithm are in the SwapMatrixRound and MatrixRound procedures respectively.
> In our work, we provide a very simple swap procedure which is easier to implement, that in turn helps us implement the SwapMatrixRound procedure, which has the same role as the MatrixRound procedure from [ACSS20] but enables us to compute PML distributions that have improved approximation guarantees. The main technical contribution of our work is summarized in Lemmas 4.4 and 4.5. Please refer to Section 4.1 for further details.
>
> {\bf Response to Question 1}:
> In brief, better approximate PML algorithms yield universal estimators for symmetric properties that are sample optimal in a broader regime of errors (see, e.g. Lines 23-60 in the introduction for details).
> To elaborate, a plug-in-based approach on any $\beta=\exp(-O(n^{1-c'}))$-approximate PML distribution (for constant $c'>0$) gives an estimator whose failure probability is roughly upper bounded by $\delta \exp(O(n^{1/3+c} + n^{1-c'}))$, equivalently a sample optimal estimator in the regime $\epsilon \gg n^{-\min(1/3,c'/2)}$ for several well known symmetric properties. Therefore, based on these conditions, asymptotically, an exact PML has the same theoretical guarantees as an $\exp(O(n^{1/3}))$-approximate PML distribution. In the revision, we will add some of the above discussion in the introduction to make the application of the approximate PML estimator more explicit.
>
> {\bf Response to Question 2}:
> Thank you for bringing this up. The discussion of why [ACSS21] only implies a universal estimator in the regime $\epsilon\gg n^{-1/4}$ appears in the introduction between Lines 32-42. As discussed in the response to question 1, any $\beta=\exp(-n^{1-c'})$-approximate PML distribution for constant $c'>0$ gives an estimator whose failure probability is upper bounded by $\delta \exp(n^{1/3+c} + n^{1-c'})$, equivalently a sample optimal estimator in the regime $\epsilon \gg n^{-\min(1/3,c'/2)}$ for several well known symmetric properties. Note that [ACSS21] gives an efficient algorithm to compute an $\exp(-\sqrt{n} \log n)$ approximate PML distribution, which in turn implies an universal estimator in the regime $\epsilon \gg n^{-\min(1/3,c'/2)}$ for $c'=1/2$, which translates to $\epsilon \gg n^{-1/4}$.
>
> In addition to the above discussion, in the results section between Lines 147-166, we have also added a comparison of our work to prior approximate PML algorithms including [ACSS21]. Also, please refer to Section 4.1 for a comparison of our work to the previous best [ACSS20] approximate PML algorithm. However, we agree that the discussion could be made clearer and in the revision, we will add further details including the discussion from the above comparing our work with [ACSS21] and also include the application of approximate PML distributions. Also, we will add additional comparisons of our work to previous works quite early in the paper.

---

### Official Review · Reviewer_BTch · 2022-07-26

**Rating:** 5
**Confidence:** 4
**Soundness:** 3 good
**Presentation:** 3 good
**Contribution:** 2 fair

**Summary:**

The paper studies the now classical method of profile maximum likelihood (PML) and provides an efficient estimator that works for tiny errors below n^{-1/4}, where n is the sample size.

**Questions:**

It would be great if the authors could address the improper definitions, references, and lemmas mentioned above.

**Limitations:**

Page 5 presents a section on the limitations of the proposed algorithm, as solving the involved convex program can be problematic in the large-sample regime. Maybe the authors can also describe some theoretical limitations, if any. I couldn't find much discussion on the potential negative social impact of the authors' work.

**Strengths And Weaknesses:**

The strength is that n^{-1/3} is one type of theoretical limit: For any plug-in estimator, a symmetric functional exists for which the plug-in estimator is suboptimal. This point is also a weakness since prior work already provided efficient estimators for any error above n^{-1/4}; closing this tiny gap between n^{-1/3} and n^{-1/4} might be incremental.

I like that the proofs are relatively short compared to prior works. However, this may rely on the fact that existing results can serve as lemmas.

It is not entirely clear what a 'sample optimal universal estimator' means in the paper. I would say it's better if the authors can provide a concrete & mathematical definition for this. Also, it doesn't seem that [ADOS16] establishes the PML estimator's optimality beyond the four mentioned functionals. And [HO19] showed that PML is sample-optimal for a "Lipschitz" functional class, which should appear around line 28.

Lemma 2.4 is simply Incorrect. Theorem 3 in [HS20] shows the existence of a sublinear estimator with no sample optimality guarantees for functionals like 1-Lipschitz. It seems that [HO19] presented something close to the desired claim.

I didn't check the proofs in the appendix. But besides the above error, looking at the references also concerned me about this paper's rigorousness. For example, the citation for [HS20] provides no more than the publishing year (how about which journal/conference?). Maybe this would hint that the paper was completed in a rush.

---

> ### Author Response · Authors · 2022-08-02
> **Response to questions and concerns raised by the reviewer**
>
> Thank you for your time and your valuable feedback. We appreciate your writing suggestions and detailed review. We respond to specific comments and concerns below.
>
> {\bf Usage of ``sample optimal universal estimators'':}
> universal estimators refer to estimators that are designed to be independent of the property of interest; in the submission, we used the term informally (thank you for pointing this out). As per your suggestion, in the revision, we will
> use clearer terminology and formally define the terms we use.
>
> {\bf\em ``The strength is that $n^{-1/3}$ is one type of theoretical limit \ldots closing this tiny gap between $n^{-1/3}$ and $n^{-1/4}$ might be incremental'':}
> Yes, the improvement in $\epsilon$ maybe be debatably small ($n^{-1/12}$) however, we think there is a big qualitative difference in achieving the worst-case information-theoretic limit of what PML can do efficiently or not. Further, we think the techniques we use to achieve it are of general theoretic interest. They simplify the matrix rounding procedure from the prior work [ACSS20] and enable both a better approximation ratio and significant running time improvements.
>
> {\bf\em ``the proofs are relatively short \ldots this may rely on the fact that existing results can serve as lemmas'':}
> we remark that the main contribution of our rounding algorithm is in proving Lemmas 4.4 and 4.5. These lemmas are non-trivial and don't follow from previous work. Though these lemmas are analogous to the MatrixRound procedure in [ACSS20], which was the main contribution of their paper, here we provide a simpler rounding algorithm which provides stronger guarantees for symmetric property estimation. We took steps to simplify the rounding algorithm and its proof and the resulting short proofs we hope are viewed as a virtue of our approach. (Please refer to the results section starting at Line 147 and also Section 4.1 starting at Line 265 for a comparison of our work to the previous approximate PML algorithms including [ACSS20].)
>
> {\bf\em ``it doesn't seem that [ADOS16] establishes the PML estimator's optimality beyond the four mentioned functionals'':}
> We agree that [ADOS16]
> does not show optimality for Lipschitz functions and we are aware that [HO19] shows optimality of PML for other properties as discussed in the related work.
> In the revision, we will make this more explicit and explain the contribution of [HO19].
>
> {\bf Incorrectness of Lemma 2.4:}
> Thank you for pointing this out!
> Indeed, we should not include Lipschitz functions in the statement of Lemma 2.4.
> Also, we agree that [HO19] presents a result which is close to what we want. In the revision, we will remove the word Lipschitz functions from the statement from Lemma 2.4 and discuss the implications of our result for other properties.
>
>
> {\bf Missing information in references:}
> Good point! We will fix this in the revision.
>
> {\bf Discussion of potentially negative social impact:}
> this work is primarily theoretical and its societal implications are limited and unclear given the scope of the submission.

---

### Official Review · Reviewer_qCnJ · 2022-07-30

**Rating:** 8
**Confidence:** 4
**Soundness:** 4 excellent
**Presentation:** 3 good
**Contribution:** 4 excellent

**Summary:**

The paper concerns the computation of an approximate profile maximum likelihood estimator (APML) -- a plug-in estimator used to estimate symmetric properties of discrete distributions. Formally, one is given access to $n$ iid samples from a discrete distribution over a set $\mathcal{D}$ and the goal is to estimate symmetric properties of this distribution (properties independent of the permutation of the labels of the distribution). This is a broad class of properties which includes practically important examples such as entropy and support size. While information theoretically optimal estimators for specific properties were known previously, the past decade or so has witnessed much interest in the development of statistically and computationally efficient _universal_ estimators for all symmetric properties. Instead of developing specialized approaches for specific properties, these approaches instead propose a unified framework for estimating all symmetric properties. Broadly speaking, there are two main approaches towards building such estimators -- one based on computing APML plug-in estimators and an alternative based on local moment matching. The local moment matching framework is both computationally and statistically efficient and is known to recover optimal sample complexity up to an accuracy threshold of $\epsilon \gg n^{-1/3}$. However, while the statistical limit of the APML approach has been established as the regime $\epsilon \approx n^{-1/3}$ with prior work demonstrating that no reasonable variant of APML succeeds below the $n^{-1/3}$ threshold, the best known computationally efficient results only hold when $\epsilon \gg n^{-1/4}$. The goal of this paper is to bridge this gap by proposing a computationally efficient estimator achieving this $n^{-1/3}$ threshold.

Technically, the APML approach first constructs a distribution over $\mathcal{D}$ from the observed data and estimates a symmetric property by evaluating the property on the fitted distribution. A naive approach to do this would be to use the distribution maximizing the likelihood of the observed sample -- i.e the empirical distribution of the samples. However, this leads to sub-optimal sample complexity. The Profile Maximum Likelihood approach instead constructs the plug-in distribution by maximizing the observed profile of the samples which consists of the multiset of frequencies observed in the sample. Unfortunately, this exact formulation while statistically more efficient and optimal in several regimes (when $\epsilon \gg n^{-1/3}$), introduces several computational barriers. Fortunately, it suffices to compute an Approximate Profile Maximum Likelihood distribution where one uses any distribution, $q$ which satisfies $\mathbb{P} (\phi, q) \geq \beta \max_p \mathbb{P} (\phi, p)$ where $\phi$ is the observed profile for $\beta \geq \exp (-n^{1-\delta})$ with $\delta > 0$ and the quality of approximation improving with $\delta$ ($\epsilon \gg n^{- \min (1/3, \delta / 2)}$). Prior work had constructed APML distributions for $\delta = 1/2$ leading to the sub-optimal $n^{-1/4}$ threshold. Informally, this approach first re-parameterizes the distribution being optimized over by discretizing it over a pre-defined set of levels (at levels $(1 + \alpha)^{i} / (n^2)$ for $i \in [0, \ell]$) and only storing the number of elements at a particular level of the distribution. Subsequently, a convex approximation to the PML objective is constructed over this re-parameterized space. Unfortunately, these two steps result in a trade-off where refining the discretization (by lowering $\alpha$) may better approximate the underlying distribution but worsens the quality of the convex relaxation. The combination of these two factors result in the optimal choice of $\alpha$ (in prior work) being $n^{-1/2}$ leading to the aforementioned sub-optimal results. The key technical idea of this paper is to effectively eliminate the first step (of approximating the true PML distribution by a discretized approximation) and instead only search for _weak_ APML distributions; i.e distributions satisfying $\mathbb{P} (\phi, q) \geq \beta \max_{p \in \Delta^R_\mathcal{D} \mathbb{P} (\phi, p)}$ where $\Delta^R_\mathcal{D}$ consists of a discretized set of distributions. Here, we are only competitive with respect to the optimal discretized distribution as opposed to the unconditionally optimal distribution. While the objective value of the recovered distribution might be a poor approximation of the optimal PML objective, this nevertheless suffices to accurately estimate a broad class of symmetric properties. In fact, the recovered distribution is at best a $\exp (- n^{-2/3})$ approximate PML distribution but nevertheless, this still suffices to achieve the optimal $\epsilon \gg n^{-1/3}$ threshold. Constructing this estimator however requires a significant refinement of the rounding algorithm which converts the fractional solution of the convex program to a discrete integral solution corresponding to a true distribution. Informally, the novel regime studied in this paper results in a solution matrix which much fewer rows than columns and the guarantee of the rounding algorithm is required to be depend exponentially in only the smaller (row number) dimension while the previous approach only achieved the much looser row + column number bound. The authors construct such a rounding algorithm leading to their improved results.

I do have concerns about the technical writing in the paper. While the proof looks correct and the technical ideas are novel, there are some steps which could benefit from further exposition and others in need of correction. Specifically, Lemmas 4.8 and 4.9 in the paper are applied in a chained fashion in the proof of Lemma 4.5 (Lemma E.1 in the appendix from line 671-677). However, as stated the precondition of the application of Lemma 4.9 are not satisfied by the conclusion of Lemma 4.8. In particular, the condition that the columns sum to integral values is not satisfied as stated in Lemma 4.8. While this is probably true since the construction of the $\mathrm{trans}$ operation (in Lemma C.1) underlying the transformation $\mathrm{PartialRound}$ is built off of the operation $\mathrm{swap}$ which maintains row and column sums by definition, this should still be stated and clarified in the main text. Likewise, this also requires a more precise formulation of Lemma C.1. Additionally, the technical exposition could also be improved surrounding the definition of the rounding algorithm in the appendix. While the main idea behind the rounding algorithm is neat and intuitive, the notation is extremely cumbersome and makes the exposition hard to follow. Providing some intuition prior to their technical definitions in Appendices C-E would help greatly.

Overall, I quite like this paper. It makes significant progress on an important problem essentially closing the gap between the best results achieved by prior computationally efficient APML based estimators and a strong information theoretic barrier. The technical ideas behind the improvement are novel and intricate and are potentially of broad interest to the theoretical machine learning and algorithms communities. While there are minor bugs in the technical statements and some of the technical writing could be improved, these seem quite easy to fix.

**** POST REBUTTAL UPDATE ****

I acknowledge the authors' response and will retain my current evaluation.

**Questions:**

It would be great if the authors could clarify some of the technical material in the paper in case my current understanding is correct.

Additionally, I'm also curious about when this $n^{-1/3}$ barrier may be circumvented for narrower classes properties as opposed to all symmetric properties. Applications of such ideas to domains beyond the discrete setting studied in this field would also be interesting to see.

**Limitations:**

Yes

**Strengths And Weaknesses:**

Strengths:

-- The paper makes strong progress on an important problem essentially achieving optimal results for this particular approach
-- The technical ideas in the paper are novel, interesting and non-trivial

Weaknesses:

-- The technical exposition in the paper could be improved both in the main text and the appendix (see main review).

---

> ### Author Response · Authors · 2022-08-02
> **Addressing the weaknesses and the questions posed by the reviewer.**
>
> Thank you for your time and your valuable feedback. We appreciate your writing suggestions and detailed review. We respond to specific points below.
>
> {\bf Concerns related to technical writing:} thank you for raising these concerns. Yes, the conclusion of Lemma 4.8 should include the following two claims: $[X 1]_i = [Y1]_i$
> and
> $[X 1]_j = [Y1]_j$
> and the preconditions of Lemma 4.9 should include
> $ [Y 1]\\_{i'} \in Z\\_{\geq 0} $
> for any
> $i' \in [i-1]$.
>
> These claims help maintain the integrality of column sums and the preconditions of Lemma 4.9. Regarding the correctness of these claims, as you noted, the above conditions are immediately implied by the swap and trans operations that preserve the row and column sums. In the revision, we will clarify the guarantees of Lemmas 4.8, 4.9 and C.1.
>
> {\bf Technical exposition of the rounding algorithm in the main text and the appendix:} thank you for your suggestions. In the revision, we will add more intuitive pictures, and try to find a clearer notation. In addition, as suggested, we will provide intuition for the technical definitions presented in Appendix C-E and other places needed.
>
> {\bf \em ``I'm also curious about when this barrier may be circumvented for narrower classes properties as opposed to all symmetric properties'':}
> you are right that the lower bound only holds when you wish to estimate all the symmetric properties. For narrower subclasses of the symmetric properties, it may be possible to circumvent the condition of $\epsilon \gg n^{-1/3}$; understanding this question is an interesting future direction (note that for specific symmetric properties some work along this direction has been carried out in [CSS10b, HO19]). We will add discussion on this question in the final revision.

---

### Meta-Review · Area_Chair_zz8y · 2022-08-30

**Recommendation:** Accept
**Confidence:** Certain

**Metareview:**

This work makes theoretical contributions to a long line of work on efficient property estimation. Specifically, the paper closes the gap on the best achievable results via the approximate profile maximum likelihood (PML) approach and the previously known information theoretic lower bound. The reviewers agreed that the technical arguments are novel and non-trivial and the result meets the bar for acceptance.

**Award:**

No

---

### Decision · Program_Chairs · 2022-09-14

Accept